# SCALABLE PRIVATE LEARNING WITH PATE

**Nicolas Papernot***
Pennsylvania State University
ngp5056@cse.psu.edu

**Shuang Song***
University of California San Diego
shs037@eng.ucsd.edu

**Ilya Mironov, Ananth Raghunathan, Kunal Talwar & Úlfar Erlingsson**
Google Brain
{mironov,pseudorandom,kunal,ulfar}@google.com

## ABSTRACT

The rapid adoption of machine learning has increased concerns about the privacy implications of machine learning models trained on sensitive data, such as medical records or other personal information. To address those concerns, one promising approach is *Private Aggregation of Teacher Ensembles*, or PATE, which transfers to a "student" model the knowledge of an ensemble of "teacher" models, with intuitive privacy provided by training teachers on disjoint data and strong privacy guaranteed by noisy aggregation of teachers' answers. However, PATE has so far been evaluated only on simple classification tasks like MNIST, leaving unclear its utility when applied to larger-scale learning tasks and real-world datasets.

In this work, we show how PATE can scale to learning tasks with large numbers of output classes and uncurated, imbalanced training data with errors. For this, we introduce new noisy aggregation mechanisms for teacher ensembles that are more selective and add less noise, and prove their tighter differential-privacy guarantees. Our new mechanisms build on two insights: the chance of teacher consensus is increased by using more concentrated noise and, lacking consensus, no answer need be given to a student. The consensus answers used are more likely to be correct, offer better intuitive privacy, and incur lower-differential privacy cost. Our evaluation shows our mechanisms improve on the original PATE on all measures, and scale to larger tasks with both high utility and very strong privacy ($\varepsilon < 1.0$).

## 1 INTRODUCTION

Many attractive applications of modern machine-learning techniques involve training models using highly sensitive data. For example, models trained on people's personal messages or detailed medical information can offer invaluable insights into real-world language usage or the diagnoses and treatment of human diseases (McMahan et al., 2017; Liu et al., 2017). A key challenge in such applications is to prevent models from revealing inappropriate details of the sensitive data—a non-trivial task, since models are known to implicitly memorize such details during training and also to inadvertently reveal them during inference (Zhang et al., 2017; Shokri et al., 2017).

Recently, two promising, new model-training approaches have offered the hope that practical, high-utility machine learning may be compatible with strong privacy-protection guarantees for sensitive training data (Abadi et al., 2017). This paper revisits one of these approaches, *Private Aggregation of Teacher Ensembles*, or PATE (Papernot et al., 2017), and develops techniques that improve its scalability and practical applicability. PATE has the advantage of being able to learn from the aggregated consensus of separate "teacher" models trained on disjoint data, in a manner that both provides intuitive privacy guarantees and is agnostic to the underlying machine-learning techniques (cf. the approach of differentially-private stochastic gradient descent (Abadi et al., 2016)). In the PATE approach multiple teachers are trained on disjoint sensitive data (e.g., different users' data), and uses the teachers' aggregate consensus answers in a black-box fashion to supervise the training of a "student" model. By publishing only the student model (keeping the teachers private) and by adding carefully-calibrated Laplacian noise to the aggregate answers used to train the student, the

---

*Equal contributions, authors ordered alphabetically. Work done while the authors were at Google Brain.

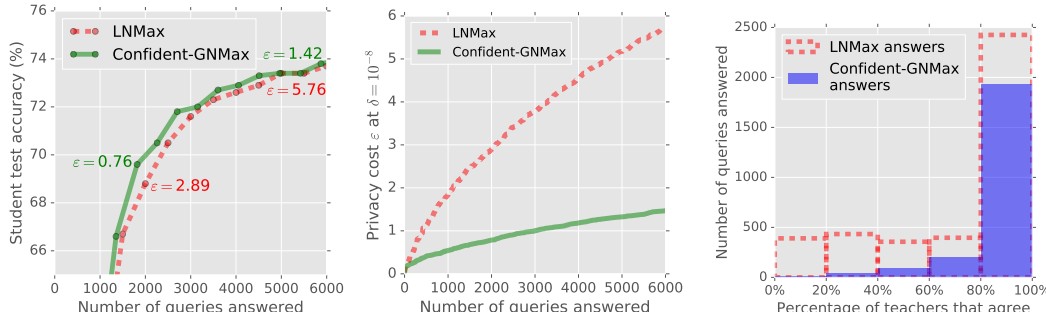

Figure 1: Our contributions are techniques (Confident-GNMax) that improve on the original PATE (LNMax) on all measures. *Left:* Accuracy is higher throughout training, despite greatly improved privacy (more in Table 1). *Middle:* The $\varepsilon$ differential-privacy bound on privacy cost is quartered, at least (more in Figure 5). *Right:* Intuitive privacy is also improved, since students are trained on answers with a much stronger consensus among the teachers (more in Figure 5). These are results for a character-recognition task, using the most favorable LNMax parameters for a fair comparison.

original PATE work showed how to establish rigorous $(\varepsilon, \delta)$ differential-privacy guarantees (Papernot et al., 2017)—a gold standard of privacy (Dwork et al., 2006). However, to date, PATE has been applied to only simple tasks, like MNIST, without any realistic, larger-scale evaluation.

The techniques presented in this paper allow PATE to be applied on a larger scale to build more accurate models, in a manner that improves both on PATE's intuitive privacy-protection due to the teachers' independent consensus as well as its differential-privacy guarantees. As shown in our experiments, the result is a gain in privacy, utility, and practicality—an uncommon joint improvement.

The primary technical contributions of this paper are new mechanisms for aggregating teachers' answers that are more selective and add less noise. On all measures, our techniques improve on the original PATE mechanism when evaluated on the same tasks using the same datasets, as described in Section 5. Furthermore, we evaluate both variants of PATE on a new, large-scale character recognition task with 150 output classes, inspired by MNIST. The results show that PATE can be successfully utilized even to uncurated datasets—with significant class imbalance as well as erroneous class labels—and that our new aggregation mechanisms improve both privacy and model accuracy.

To be more selective, our new mechanisms leverage some pleasant synergies between privacy and utility in PATE aggregation. For example, when teachers disagree, and there is no real consensus, the privacy cost is much higher; however, since such disagreement also suggest that the teachers may not give a correct answer, the answer may simply be omitted. Similarly, teachers may avoid giving an answer where the student already is confidently predicting the right answer. Additionally, we ensure that these selection steps are themselves done in a private manner.

To add less noise, our new PATE aggregation mechanisms sample Gaussian noise, since the tails of that distribution diminish far more rapidly than those of the Laplacian noise used in the original PATE work. This reduction greatly increases the chance that the noisy aggregation of teachers' votes results in the correct consensus answer, which is especially important when PATE is scaled to learning tasks with large numbers of output classes. However, changing the sampled noise requires redoing the entire PATE privacy analysis from scratch (see Section 4 and details in Appendix A).

Finally, of independent interest are the details of our evaluation extending that of the original PATE work. In particular, we find that the virtual adversarial training (VAT) technique of Miyato et al. (2017) is a good basis for semi-supervised learning on tasks with many classes, outperforming the improved GANs by Salimans et al. (2016) used in the original PATE work. Furthermore, we explain how to tune the PATE approach to achieve very strong privacy ($\varepsilon \approx 1.0$) along with high utility, for our real-world character recognition learning task.

This paper is structured as follows: Section 2 is the related work section; Section 3 gives a background on PATE and an overview of our work; Section 4 describes our improved aggregation mechanisms; Section 5 details our experimental evaluation; Section 6 offers conclusions; and proofs are deferred to the Appendices.

## 2 RELATED WORK

Differential privacy is by now the gold standard of privacy. It offers a rigorous framework whose threat model makes few assumptions about the adversary's capabilities, allowing differentially private algorithms to effectively cope against strong adversaries. This is not the case of all privacy definitions, as demonstrated by successful attacks against anonymization techniques (Aggarwal, 2005; Narayanan & Shmatikov, 2008; Bindschaedler et al., 2017).

The first learning algorithms adapted to provide differential privacy with respect to their training data were often linear and convex (Pathak et al., 2010; Chaudhuri et al., 2011; Song et al., 2013; Bassily et al., 2014; Hamm et al., 2016). More recently, successful developments in deep learning called for differentially private stochastic gradient descent algorithms (Abadi et al., 2016), some of which have been tailored to learn in federated (McMahan et al., 2017) settings.

Differentially private selection mechanisms like GNMax (Section 4.1) are commonly used in hypothesis testing, frequent itemset mining, and as building blocks of more complicated private mechanisms. The most commonly used differentially private selection mechanisms are exponential mechanism (McSherry & Talwar, 2007) and LNMax (Bhaskar et al., 2010). Recent works offer lower bounds on sample complexity of such problem (Steinke & Ullman, 2017; Bafna & Ullman, 2017).

The Confident and Interactive Aggregator proposed in our work (Section 4.2 and Section 4.3 resp.) use the intuition that selecting samples under certain constraints could result in better training than using samples uniformly at random. In Machine Learning Theory, active learning (Cohn et al., 1994) has been shown to allow learning from fewer labeled examples than the passive case (see e.g. Hanneke (2014)). Similarly, in model stealing (Tramèr et al., 2016), a goal is to learn a model from limited access to a teacher network. There is previous work in differential privacy literature (Hardt & Rothblum, 2010; Roth & Roughgarden, 2010) where the mechanism first *decides* whether or not to answer a query, and then privately answers the queries it chooses to answer using a traditional noise-addition mechanism. In these cases, the sparse vector technique (Dwork & Roth, 2014, Chapter 3.6) helps bound the privacy cost in terms of the number of answered queries. This is in contrast to our work where a constant *fraction* of queries get answered and the sparse vector technique does not seem to help reduce the privacy cost. Closer to our work, Bun et al. (2017) consider a setting where the answer to a query of interest is often either very large or very small. They show that a sparse vector-like analysis applies in this case, where one pays only for queries that are in the middle.

## 3 BACKGROUND AND OVERVIEW

We introduce essential components of our approach towards a generic and flexible framework for machine learning with provable privacy guarantees for training data.

### 3.1 THE PATE FRAMEWORK

Here, we provide an overview of the PATE framework. To protect the privacy of training data during learning, PATE transfers knowledge from an ensemble of teacher models trained on partitions of the data to a student model. Privacy guarantees may be understood intuitively and expressed rigorously in terms of differential privacy.

Illustrated in Figure 2, the PATE framework consists of three key parts: (1) an ensemble of $n$ teacher models, (2) an aggregation mechanism and (3) a student model.

**Teacher models:** Each teacher is a model trained independently on a subset of the data whose privacy one wishes to protect. The data is partitioned to ensure no pair of teachers will have trained on overlapping data. Any learning technique suitable for the data can be used for any teacher. Training each teacher on a *partition* of the sensitive data produces $n$ different models solving the same task. At inference, teachers independently predict labels.

**Aggregation mechanism:** When there is a strong consensus among teachers, the label they almost all agree on does not depend on the model learned by any given teacher. Hence, this collective decision is intuitively private with respect to any given training point—because such a point could have been included only in one of the teachers' training set. To provide rigorous guarantees of differential privacy, the aggregation mechanism of the original PATE framework counts votes assigned

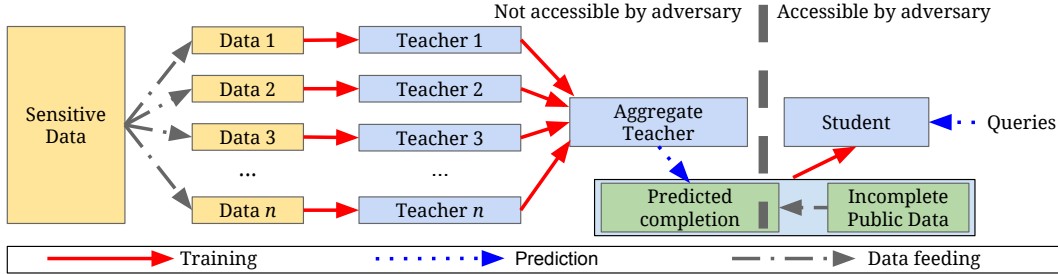

Figure 2: Overview of the approach: (1) an ensemble of teachers is trained on disjoint subsets of the sensitive data, (2) a student model is trained on public data labeled using the ensemble.

to each class, adds carefully calibrated Laplacian noise to the resulting vote histogram, and outputs the class with the most noisy votes as the ensemble's prediction. This mechanism is referred to as the max-of-Laplacian mechanism, or LNMax, going forward.

For samples $x$ and classes $1, \ldots, m$, let $f_j(x) \in [m]$ denote the $j$-th teacher model's prediction and $n_i$ denote the vote count for the $i$-th class (i.e., $n_i \triangleq |f_j(x) = i|$). The output of the mechanism is $\mathcal{A}(x) \triangleq \mathrm{argmax}_i \left( n_i(x) + \mathrm{Lap}\left(1/\gamma\right) \right)$. Through a rigorous analysis of this mechanism, the PATE framework provides a differentially private API: the privacy cost of each aggregated prediction made by the teacher ensemble is known.

**Student model:** PATE's final step involves the training of a student model by knowledge transfer from the teacher ensemble using access to public—but *unlabeled*—data. To limit the privacy cost of labeling them, queries are only made to the aggregation mechanism for a subset of public data to train the student in a semi-supervised way using a fixed number of queries. The authors note that every additional ensemble prediction increases the privacy cost spent and thus cannot work with unbounded queries. Fixed queries fixes privacy costs as well as diminishes the value of attacks analyzing model parameters to recover training data (Zhang et al., 2017). The student only sees public data and privacy-preserving labels.

## 3.2 DIFFERENTIAL PRIVACY

Differential privacy (Dwork et al., 2006) requires that the sensitivity of the distribution of an algorithm's output to small perturbations of its input be limited. The following variant of the definition captures this intuition formally:

**Definition 1.** *A randomized mechanism $\mathcal{M}$ with domain $\mathcal{D}$ and range $\mathcal{R}$ satisfies $(\varepsilon, \delta)$-differential privacy if for any two adjacent inputs $D, D' \in \mathcal{D}$ and for any subset of outputs $S \subseteq \mathcal{R}$ it holds that:*

$$\mathbf{Pr}[\mathcal{M}(D) \in S] \le e^\varepsilon \cdot \mathbf{Pr}[\mathcal{M}(D') \in S] + \delta. \tag{1}$$

For our application of differential privacy to ML, adjacent inputs are defined as two datasets that only differ by one training example and the randomized mechanism $\mathcal{M}$ would be the model training algorithm. The privacy parameters have the following natural interpretation: $\varepsilon$ is an upper bound on the loss of privacy, and $\delta$ is the probability with which this guarantee may not hold. Composition theorems (Dwork & Roth, 2014) allow us to keep track of the privacy cost when we run a sequence of mechanisms.

## 3.3 RÉNYI DIFFERENTIAL PRIVACY

Papernot et al. (2017) note that the natural approach to bounding PATE's privacy loss—by bounding the privacy cost of each label queried and using strong composition (Dwork et al., 2010) to derive the total cost—yields loose privacy guarantees. Instead, their approach uses *data-dependent* privacy analysis. This takes advantage of the fact that when the consensus among the teachers is very strong, the plurality outcome has overwhelming likelihood leading to a very small privacy cost whenever the consensus occurs. To capture this effect quantitatively, Papernot et al. (2017) rely on the *moments*

*accountant*, introduced by Abadi et al. (2016) and building on previous work (Bun & Steinke, 2016; Dwork & Rothblum, 2016).

In this section, we recall the language of Rényi Differential Privacy or RDP (Mironov, 2017). RDP generalizes pure differential privacy ($\delta = 0$) and is closely related to the moments accountant. We choose to use RDP as a more natural analysis framework when dealing with our mechanisms that use Gaussian noise. Defined below, the RDP of a mechanism is stated in terms of the Rényi divergence.

**Definition 2** (Rényi Divergence). *The Rényi divergence of order $\lambda$ between two distributions $P$ and $Q$ is defined as:*

$$D_\lambda(P\|Q) \triangleq \frac{1}{\lambda - 1} \log \mathbb{E}_{x \sim Q}\left[(P(x)/Q(x))^\lambda\right] = \frac{1}{\lambda - 1} \log \mathbb{E}_{x \sim P}\left[(P(x)/Q(x))^{\lambda - 1}\right].$$

**Definition 3** (Rényi Differential Privacy (RDP)). *A randomized mechanism $\mathcal{M}$ is said to guarantee $(\lambda, \varepsilon)$-RDP with $\lambda \geq 1$ if for any neighboring datasets $D$ and $D'$,*

$$D_\lambda(\mathcal{M}(D)\|\mathcal{M}(D')) = \frac{1}{\lambda - 1} \log \mathbb{E}_{x \sim \mathcal{M}(D)}\left[\left(\frac{\mathbf{Pr}\left[\mathcal{M}(D) = x\right]}{\mathbf{Pr}\left[\mathcal{M}(D') = x\right]}\right)^{\lambda - 1}\right] \leq \varepsilon.$$

RDP generalizes pure differential privacy in the sense that $\varepsilon$-differential privacy is equivalent to $(\infty, \varepsilon)$-RDP. Mironov (2017) proves the following key facts that allow easy composition of RDP guarantees and their conversion to $(\varepsilon, \delta)$-differential privacy bounds.

**Theorem 4** (Composition). *If a mechanism $\mathcal{M}$ consists of a sequence of adaptive mechanisms $\mathcal{M}_1, \ldots, \mathcal{M}_k$ such that for any $i \in [k]$, $\mathcal{M}_i$ guarantees $(\lambda, \varepsilon_i)$-RDP, then $\mathcal{M}$ guarantees $(\lambda, \sum_{i=1}^{k} \varepsilon_i)$-RDP.*

**Theorem 5** (From RDP to DP). *If a mechanism $\mathcal{M}$ guarantees $(\lambda, \varepsilon)$-RDP, then $\mathcal{M}$ guarantees $(\varepsilon + \frac{\log 1/\delta}{\lambda - 1}, \delta)$-differential privacy for any $\delta \in (0, 1)$.*

While both $(\varepsilon, \delta)$-differential privacy and RDP are relaxations of pure $\varepsilon$-differential privacy, the two main advantages of RDP are as follows. First, it composes nicely; second, it captures the privacy guarantee of Gaussian noise in a much cleaner manner compared to $(\varepsilon, \delta)$-differential privacy. This lets us do a careful privacy analysis of the GNMax mechanism as stated in Theorem 6. While the analysis of Papernot et al. (2017) leverages the first aspect of such frameworks with the Laplace noise (LNMax mechanism), our analysis of the GNMax mechanism relies on both.

### 3.4 PATE Aggregation Mechanisms

The aggregation step is a crucial component of PATE. It enables knowledge transfer from the teachers to the student while enforcing privacy. We improve the LNMax mechanism used by Papernot et al. (2017) which adds Laplace noise to teacher votes and outputs the class with the highest votes.

First, we add Gaussian noise with an accompanying privacy analysis in the RDP framework. This modification effectively reduces the noise needed to achieve the same privacy cost per student query.

Second, the aggregation mechanism is now *selective*: teacher votes are analyzed to decide which student queries are *worth* answering. This takes into account both the privacy cost of each query and its payout in improving the student's utility. Surprisingly, our analysis shows that these two metrics are not at odds and in fact align with each other: the privacy cost is the smallest when teachers agree, and when teachers agree, the label is more likely to be correct thus being more useful to the student.

Third, we propose and study an *interactive* mechanism that takes into account not only teacher votes on a queried example but possible student predictions on that query. Now, queries worth answering are those where the teachers agree on a class but the student is not confident in its prediction on that class. This third modification aligns the two metrics discussed above even further: queries where the student already agrees with the consensus of teachers are not worth expending our privacy budget on, but queries where the student is less confident are useful and answered at a small privacy cost.

### 3.5 DATA-DEPENDENT PRIVACY IN PATE

A direct privacy analysis of the aggregation mechanism, for reasonable values of the noise parameter, allows answering only few queries before the privacy cost becomes prohibitive. The original PATE proposal used a data-dependent analysis, exploiting the fact that when the teachers have large agreement, the privacy cost is usually much smaller than the data-independent bound would suggest.

In our work, we perform a data-dependent privacy analysis of the aggregation mechanism with Gaussian noise. This change of noise distribution turns out be technically much more challenging than the Laplace noise case and we defer the details to Appendix A. This increased complexity of the analysis however does not make the algorithm any more complicated and thus allows us to improve the privacy-utility tradeoff.

**Sanitizing the privacy cost via smooth sensitivity analysis.** An additional challenge with data-dependent privacy analyses arises from the fact that the privacy cost itself is now a function of the private data. Further, the data-dependent bound on the privacy cost has large global sensitivity (a metric used in differential privacy to calibrate the noise injected) and is therefore difficult to sanitize. To remedy this, we use the smooth sensitivity framework proposed by Nissim et al. (2007).

Appendix B describes how we add noise to the computed privacy cost using this framework to publish a sanitized version of the privacy cost. Section B.1 defines smooth sensitivity and outlines algorithms 3–5 that compute it. The rest of Appendix B argues the correctness of these algorithms. The final analysis shows that the incremental cost of sanitizing our privacy estimates is modest— less than 50% of the raw estimates—thus enabling us to use precise data-dependent privacy analysis while taking into account its privacy implications.

## 4 IMPROVED AGGREGATION MECHANISMS FOR PATE

The privacy guarantees provided by PATE stem from the design and analysis of the aggregation step. Here, we detail our improvements to the mechanism used by Papernot et al. (2017). As outlined in Section 3.4, we first replace the Laplace noise added to teacher votes with Gaussian noise, adapting the data-dependent privacy analysis. Next, we describe the Confident and Interactive Aggregators that select queries worth answering in a privacy-preserving way: the privacy budget is shared between the query selection and answer computation. The aggregators use different heuristics to select queries: the former does not take into account student predictions, while the latter does.

### 4.1 THE GNMAX AGGREGATOR AND ITS PRIVACY GUARANTEE

This section uses the following notation. For a sample $x$ and classes 1 to $m$, let $f_j(x) \in [m]$ denote the $j$-th teacher model's prediction on $x$ and $n_i(x)$ denote the vote count for the $i$-th class (i.e., $n_i(x) = |\{j \colon f_j(x) = i\}|$). We define a Gaussian NoisyMax (GNMax) aggregation mechanism as:

$$\mathcal{M}_\sigma(x) \triangleq \operatorname*{argmax}_i \left\{ n_i(x) + \mathcal{N}(0, \sigma^2) \right\},$$

where $\mathcal{N}(0, \sigma^2)$ is the Gaussian distribution with mean 0 and variance $\sigma^2$. The aggregator outputs the class with noisy plurality after adding Gaussian noise to each vote count. In what follow, *plurality* more generally refers to the highest number of teacher votes assigned among the classes.

The Gaussian distribution is more concentrated than the Laplace distribution used by Papernot et al. (2017). This concentration directly improves the aggregation's utility when the number of classes $m$ is large. The GNMax mechanism satisfies $(\lambda, \lambda/\sigma^2)$-RDP, which holds for all inputs and all $\lambda \geq 1$ (precise statements and proofs of claims in this section are deferred to Appendix A). A straightforward application of composition theorems leads to loose privacy bounds. As an example, the standard advanced composition theorem applied to experiments in the last two rows of Table 1 would give us $\varepsilon = 8.42$ and $\varepsilon = 10.14$ resp. at $\delta = 10^{-8}$ for the Glyph dataset.

To refine these, we work out a careful *data-dependent* analysis that yields values of $\varepsilon$ smaller than 1 for the same $\delta$. The following theorem translates data-independent RDP guarantees for higher orders into a data-dependent RDP guarantee for a smaller order $\lambda$. We use it in conjunction with Proposition 7 to bound the privacy cost of each query to the GNMax algorithm as a function of $\tilde{q}$, the probability that the most common answer will not be output by the mechanism.

**Theorem 6** (informal). *Let $\mathcal{M}$ be a randomized algorithm with $(\mu_1, \varepsilon_1)$-RDP and $(\mu_2, \varepsilon_2)$-RDP guarantees and suppose that given a dataset $D$, there exists a likely outcome $i^*$ such that $\mathbf{Pr}\left[\mathcal{M}(D) \neq i^*\right] \leq \tilde{q}$. Then the data-dependent Rényi differential privacy for $\mathcal{M}$ of order $\lambda \leq \mu_1, \mu_2$ at $D$ is bounded by a function of $\tilde{q}, \mu_1, \varepsilon_1, \mu_2, \varepsilon_2$, which approaches 0 as $\tilde{q} \to 0$.*

The new bound improves on the data-independent privacy for $\lambda$ as long as the distribution of the algorithm's output *on that input* has a strong peak (i.e., $\tilde{q} \ll 1$). Values of $\tilde{q}$ close to 1 could result in a looser bound. Therefore, in practice we take the minimum between this bound and $\lambda/\sigma^2$ (the data-independent one). The theorem generalizes Theorem 3 from Papernot et al. (2017), where it was shown for a mechanism satisfying $\varepsilon$-differential privacy (i.e., $\mu_1 = \mu_2 = \infty$ and $\varepsilon_1 = \varepsilon_2$).

The final step in our analysis uses the following lemma to bound the probability $\tilde{q}$ when $i^*$ corresponds to the class with the true plurality of teacher votes.

**Proposition 7.** *For any $i^* \in [m]$, we have $\mathbf{Pr}\left[\mathcal{M}_\sigma(D) \neq i^*\right] \leq \frac{1}{2} \sum_{i \neq i^*} \mathrm{erfc}\left(\frac{n_{i^*} - n_i}{2\sigma}\right)$, where* $\mathrm{erfc}$ *is the complementary error function.*

In Appendix A, we detail how these results translate to privacy bounds. In short, for each query to the GNMax aggregator, given teacher votes $n_i$ and the class $i^*$ with maximal support, Proposition 7 gives us the value of $\tilde{q}$ to use in Theorem 6. We optimize over $\mu_1$ and $\mu_2$ to get a data-dependent RDP guarantee for any order $\lambda$. Finally, we use composition properties of RDP to analyze a sequence of queries, and translate the RDP bound back to an $(\varepsilon, \delta)$-DP bound.

**Expensive queries.** This data-dependent privacy analysis leads us to the concept of an *expensive* query in terms of its privacy cost. When teacher votes largely disagree, some $n_{i^*} - n_i$ values may be small leading to a large value for $\tilde{q}$: i.e., the lack of consensus amongst teachers indicates that the aggregator is likely to output a wrong label. Thus expensive queries from a privacy perspective are often bad for training too. Conversely, queries with strong consensus enable tight privacy bounds. This synergy motivates the aggregation mechanisms discussed in the following sections: they evaluate the strength of the consensus before answering a query.

## 4.2 THE CONFIDENT-GNMAX AGGREGATOR

In this section, we propose a refinement of the GNMax aggregator that enables us to filter out queries for which teachers do not have a sufficiently strong consensus. This filtering enables the teachers to avoid answering expensive queries. We also take note to do this selection step itself in a private manner.

The proposed *Confident Aggregator* is described in Algorithm 1. To select queries with overwhelming consensus, the algorithm checks if the plurality vote crosses a threshold $T$. To enforce privacy in this step, the comparison is done after adding Gaussian noise with variance $\sigma_1^2$. Then, for queries that pass this noisy threshold check, the aggregator proceeds with the usual GNMax mechanism with a smaller variance $\sigma_2^2$. For queries that do not pass the noisy threshold check, the aggregator simply returns $\perp$ and the student discards this example in its training.

In practice, we often choose significantly higher values for $\sigma_1$ compared to $\sigma_2$. This is because we pay the cost of the noisy threshold check *always*, and without the benefit of knowing that the consensus is strong. We pick $T$ so that queries where the plurality gets less than half the votes (often very expensive) are unlikely to pass the threshold after adding noise, but we still have a high enough yield amongst the queries with a strong consensus. This tradeoff leads us to look for $T$'s between $0.6\times$ to $0.8\times$ the number of teachers.

The privacy cost of this aggregator is intuitive: we pay for the threshold check for every query, and for the GNMax step only for queries that pass the check. In the work of Papernot et al. (2017), the mechanism paid a privacy cost for every query, expensive or otherwise. In comparison, the Confident Aggregator expends a much smaller privacy cost to check against the threshold, and by answering a significantly smaller fraction of expensive queries, it expends a lower privacy cost overall.

## 4.3 THE INTERACTIVE-GNMAX AGGREGATOR

While the Confident Aggregator excludes expensive queries, it ignores the possibility that the student might receive labels that contribute little to learning, and in turn to its utility. By incorporating the

---

**Algorithm 1 – Confident-GNMax Aggregator:** given a query, consensus among teachers is first estimated in a privacy-preserving way to then only reveal confident teacher predictions.

---

**Input:** input $x$, threshold $T$, noise parameters $\sigma_1$ and $\sigma_2$
1: **if** $\max_i\{n_j(x)\} + \mathcal{N}(0, \sigma_1^2) \geq T$ **then**  $\qquad\qquad$ ▷ Privately check for consensus
2: $\qquad$ **return** $\operatorname{argmax}_j \left\{n_j(x) + \mathcal{N}(0, \sigma_2^2)\right\}$  $\qquad$ ▷ Run the usual max-of-Gaussian
3: **else**
4: $\qquad$ **return** $\perp$
5: **end if**

---

**Algorithm 2 – Interactive-GNMax Aggregator**: the protocol first compares student predictions to the teacher votes in a privacy-preserving way to then either (a) reinforce the student prediction for the given query or (b) provide the student with a new label predicted by the teachers.

---

**Input:** input $x$, confidence $\gamma$, threshold $T$, noise parameters $\sigma_1$ and $\sigma_2$, total number of teachers $M$
1: Ask the student to provide prediction scores $\mathbf{p}(x)$
2: **if** $\max_j\{n_j(x) - Mp_j(x)\} + \mathcal{N}(0, \sigma_1^2) \geq T$ **then**  $\qquad$ ▷ Student does not agree with teachers
3: $\qquad$ **return** $\operatorname{argmax}_j\{n_j(x) + \mathcal{N}(0, \sigma_2^2)\}$  $\qquad\qquad$ ▷ Teachers provide new label
4: **else if** $\max\{p_i(x)\} > \gamma$ **then**  $\qquad\qquad$ ▷ Student agrees with teachers and is confident
5: $\qquad$ **return** $\arg\max_j p_j(x)$  $\qquad\qquad\qquad\qquad$ ▷ Reinforce student's prediction
6: **else**
7: $\qquad$ **return** $\perp$  $\qquad\qquad\qquad\qquad\qquad\qquad$ ▷ No output given for this label
8: **end if**

---

student's current predictions for its public training data, we design an *Interactive Aggregator* that discards queries where the student already confidently predicts the same label as the teachers.

Given a set of queries, the Interactive Aggregator (Algorithm 2) selects those answered by comparing student predictions to teacher votes for each class. Similar to Step 1 in the Confident Aggregator, queries where the plurality of these noised differences crosses a threshold are answered with GN-Max. This noisy threshold suffices to enforce privacy of the first step because student predictions can be considered public information (the student is trained in a differentially private manner).

For queries that fail this check, the mechanism reinforces the predicted student label if the student is confident enough and does this without looking at teacher votes again. This limited form of supervision comes at a small privacy cost. Moreover, the order of the checks ensures that a student falsely confident in its predictions on a query is not accidentally reinforced if it disagrees with the teacher consensus. The privacy accounting is identical to the Confident Aggregator except in considering the difference between teachers and the student instead of only the teachers votes.

In practice, the Confident Aggregator can be used to start training a student when it can make no meaningful predictions and training can be finished off with the Interactive Aggregator after the student gains some proficiency.

## 5 EXPERIMENTAL EVALUATION

Our goal is first to show that the improved aggregators introduced in Section 4 enable the application of PATE to uncurated data, thus departing from previous results on tasks with balanced and well-separated classes. We experiment with the Glyph dataset described below to address two aspects left open by Papernot et al. (2017): (a) the performance of PATE on a task with a larger number of classes (the framework was only evaluated on datasets with at most 10 classes) and (b) the privacy-utility tradeoffs offered by PATE on data that is class imbalanced and partly mislabeled. In Section 5.2, we evaluate the improvements given by the GNMax aggregator over its Laplace counterpart (LNMax) and demonstrate the necessity of the Gaussian mechanism for uncurated tasks.

In Section 5.3, we then evaluate the performance of PATE with both the Confident and Interactive Aggregators on all datasets used to benchmark the original PATE framework, in addition to Glyph. With the right teacher and student training, the two mechanisms from Section 4 achieve high accuracy with very tight privacy bounds. Not answering queries for which teacher consensus is too

low (Confident-GNMax) or the student's predictions already agree with teacher votes (Interactive-GNMax) better aligns utility and privacy: queries are answered at a significantly reduced cost.

## 5.1 Experimental Setup

**MNIST, SVHN, and the UCI Adult databases.** We evaluate with two computer vision tasks (MNIST and Street View House Numbers (Netzer et al., 2011)) and census data from the UCI Adult dataset (Kohavi, 1996). This enables a comparative analysis of the utility-privacy tradeoff achieved with our Confident-GNMax aggregator and the LNMax originally used in PATE. We replicate the experimental setup and results found in Papernot et al. (2017) with code and teacher votes made available online. The source code for the privacy analysis in this paper as well as supporting data required to run this analysis is available on Github.[1]

A detailed description of the experimental setup can be found in Papernot et al. (2017); we provide here only a brief overview. For MNIST and SVHN, teachers are convolutional networks trained on partitions of the training set. For UCI Adult, each teacher is a random forest. The test set is split in two halves: the first is used as unlabeled inputs to simulate the student's public data and the second is used as a hold out to evaluate test performance. The MNIST and SVHN students are convolutional networks trained using semi-supervised learning with GANs à la Salimans et al. (2016). The student for the Adult dataset are fully supervised random forests.

**Glyph.** This optical character recognition task has *an order of magnitude more classes* than all previous applications of PATE. The Glyph dataset also possesses many characteristics shared by real-world tasks: e.g., it is imbalanced and some inputs are mislabeled. Each input is a $28 \times 28$ grayscale image containing a single glyph generated synthetically from a collection of over 500K computer fonts.[2] Samples representative of the difficulties raised by the data are depicted in Figure 3. The task is to classify inputs as one of the 150 Unicode symbols used to generate them.

This set of 150 classes results from pre-processing efforts. We discarded additional classes that had few samples; some classes had at least 50 times fewer inputs than the most popular classes, and these were almost exclusively incorrectly labeled inputs. We also merged classes that were too ambiguous for even a human to differentiate them. Nevertheless, a manual inspection of samples grouped by classes—favorably to the human observer—led to the conservative estimate that some classes remain 5 times more frequent, and mislabeled inputs represent at least $10\%$ of the data.

To simulate the availability of private and public data (see Section 3.1), we split data originally marked as the training set (about 65M points) into partitions given to the teachers. Each teacher is a ResNet (He et al., 2016) made of 32 leaky ReLU layers. We train on batches of 100 inputs for 40K steps using SGD with momentum. The learning rate, initially set to 0.1, is decayed after 10K steps to 0.01 and again after 20K steps to 0.001. These parameters were found with a grid search.

We split holdout data in two subsets of 100K and 400K samples: the first acts as public data to train the student and the second as its testing data. The student architecture is a convolutional network learnt in a semi-supervised fashion with virtual adversarial training (VAT) from Miyato et al. (2017). Using unlabeled data, we show how VAT can regularize the student by making predictions constant in *adversarial*[3] directions. Indeed, we found that GANs did not yield as much utility for Glyph as for MNIST or SVHN. We train with Adam for 400 epochs and a learning rate of $6 \cdot 10^{-5}$.

## 5.2 Comparing the LNMax and GNMax Mechanisms

Section 4.1 introduces the GNMax mechanism and the accompanying privacy analysis. With a Gaussian distribution, whose tail diminishes more rapidly than the Laplace distribution, we expect better utility when using the new mechanism (albeit with a more involved privacy analysis).

To study the tradeoff between privacy and accuracy with the two mechanisms, we run experiments training several ensembles of $M$ teachers for $M \in \{100, 500, 1000, 5000\}$ on the Glyph data. Re-

---

[1] `https://github.com/tensorflow/models/tree/master/research/differential_privacy`

[2] Glyph data is not public but similar data is available publicly as part of the notMNIST dataset.

[3] In this context, the adversarial component refers to the phenomenon commonly referred to as adversarial examples (Biggio et al., 2013; Szegedy et al., 2014) and not to the adversarial training approach taken in GANs.

call that 65 million training inputs are partitioned and distributed among the $M$ teachers with each teacher receiving between 650K and 13K inputs for the values of $M$ above. The test data is used to query the teacher ensemble and the resulting labels (after the LNMax and GNMax mechanisms) are compared with the ground truth labels provided in the dataset. This predictive performance of the teachers is essential to good student training with accurate labels and is a useful proxy for utility.

For each mechanism, we compute $(\varepsilon, \delta)$-differential privacy guarantees. As is common in literature, for a dataset on the order of $10^8$ samples, we choose $\delta = 10^{-8}$ and denote the corresponding $\varepsilon$ as the privacy cost. The total $\varepsilon$ is calculated on a subset of 4,000 queries, which is representative of the number of labels needed by a student for accurate training (see Section 5.3). We visualize in Figure 4 the effect of the noise distribution (left) and the number of teachers (right) on the tradeoff between privacy costs and label accuracy.

**Observations.** On the left of Figure 1, we compare our GNMax aggregator to the LNMax aggregator used by the original PATE proposal, on an ensemble of 1000 teachers and for varying noise scales $\sigma$. At fixed test accuracy, the GNMax algorithm consistently outperforms the LNMax mechanism in terms of privacy cost. To explain this improved performance, recall notation from Section 4.1. For both mechanisms, the data dependent privacy cost scales linearly with $\tilde{q}$—the likelihood of an answer other than the true plurality. The value of $\tilde{q}$ falls of as $\exp(-x^2)$ for GNMax and $\exp(-x)$ for LNMax, where $x$ is the ratio $(n_{i^*} - n_i)/\sigma$. Thus, when $n_{i^*} - n_i$ is (say) $4\sigma$, LNMax would have $\tilde{q} \approx e^{-4} = 0.018...$, whereas GNMax would have $\tilde{q} \approx e^{-16} \approx 10^{-7}$, thereby leading to a much higher likelihood of returning the true plurality. Moreover, this reduced $\tilde{q}$ translates to a smaller privacy cost for a given $\sigma$ leading to a better utility-privacy tradeoff.

As long as each teacher has sufficient data to learn a good-enough model, increasing the number $M$ of teachers improves the tradeoff—as illustrated on the right of Figure 4 with GNMax. The larger ensembles lower the privacy cost of answering queries by tolerating larger $\sigma$'s. Combining the two observations made in this Figure, for a fixed label accuracy, we lower privacy costs by switching to the GNMax aggregator and training a larger number $M$ of teachers.

## 5.3 STUDENT TRAINING WITH THE GNMAX AGGREGATION MECHANISMS

As outlined in Section 3, we train a student on public data labeled by the aggregation mechanisms. We take advantage of PATE's flexibility and apply the technique that performs best on each dataset: semi-supervised learning with Generative Adversarial Networks (Salimans et al., 2016) for MNIST and SVHN, Virtual Adversarial Training (Miyato et al., 2017) for Glyph, and fully-supervised random forests for UCI Adult. In addition to evaluating the total privacy cost associated with training the student model, we compare its utility to a non-private baseline obtained by training on the sensitive data (used to train teachers in PATE): we use the baselines of 99.2%, 92.8%, and 85.0% reported by Papernot et al. (2017) respectively for MNIST, SVHN, and UCI Adult, and we measure a baseline of 82.2% for Glyph. We compute $(\varepsilon, \delta)$-privacy bounds and denote the privacy cost as the $\varepsilon$ value at a value of $\delta$ set accordingly to number of training samples.

**Confident-GNMax Aggregator.** Given a pool of 500 to 12,000 samples to learn from (depending on the dataset), the student submits queries to the teacher ensemble running the Confident-GNMax aggregator from Section 4.2. A grid search over a range of plausible values for parameters $T$, $\sigma_1$ and $\sigma_2$ yielded the values reported in Table 1, illustrating the tradeoff between utility and privacy achieved. We additionally measure the number of queries selected by the teachers to be answered and compare student utility to a non-private baseline.

The Confident-GNMax aggregator outperforms LNMax for the four datasets considered in the original PATE proposal: it reduces the privacy cost $\varepsilon$, increases student accuracy, or both simultaneously. On the uncurated Glyph data, despite the imbalance of classes and mislabeled data (as evidenced by the 82.2% baseline), the Confident Aggregator achieves 73.5% accuracy with a privacy cost of just $\varepsilon = 1.02$. Roughly 1,300 out of 12,000 queries made are not answered, indicating that several expensive queries were successfully avoided. This selectivity is analyzed in more details in Section 5.4.

**Interactive-GNMax Aggregator.** On Glyph, we evaluate the utility and privacy of an interactive training routine that proceeds in *two rounds*. Round one runs student training with a Confident

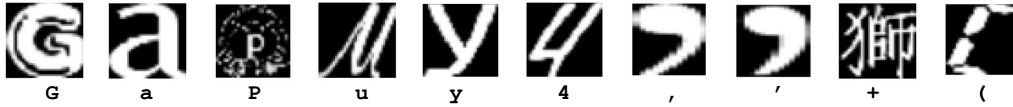

Figure 3: **Some example inputs from the Glyph dataset along with the class they are labeled as.** Note the ambiguity (between the comma and apostrophe) and the mislabeled input.

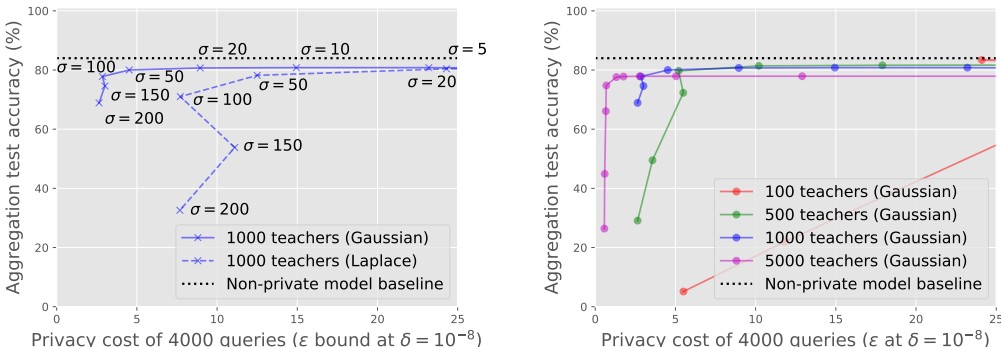

Figure 4: **Tradeoff between utility and privacy for the LNMax and GNMax aggregators on Glyph:** effect of the noise distribution (left) and size of the teacher ensemble (right). The LNMax aggregator uses a Laplace distribution and GNMax a Gaussian. Smaller values of the privacy cost $\varepsilon$ (often obtained by increasing the noise scale $\sigma$—see Section 4) and higher accuracy are better.

| Dataset | Aggregator | Queries answered | Privacy bound $\varepsilon$ | Accuracy Student | Baseline |
|---------|-----------|------------------|------------------|---------|----------|
| MNIST | LNMax (Papernot et al., 2017) | 100 | 2.04 | 98.0% | 99.2% |
| | LNMax (Papernot et al., 2017) | 1,000 | 8.03 | 98.1% | |
| | Confident-GNMax ($T$=200, $\sigma_1$=150, $\sigma_2$=40) | 286 | **1.97** | **98.5%** | |
| SVHN | LNMax (Papernot et al., 2017) | 500 | 5.04 | 82.7% | 92.8% |
| | LNMax (Papernot et al., 2017) | 1,000 | 8.19 | 90.7% | |
| | Confident-GNMax ($T$=300, $\sigma_1$=200, $\sigma_2$=40) | 3,098 | **4.96** | **91.6%** | |
| Adult | LNMax (Papernot et al., 2017) | 500 | 2.66 | 83.0% | 85.0% |
| | Confident-GNMax ($T$=300, $\sigma_1$=200, $\sigma_2$=40) | 524 | **1.90** | **83.7%** | |
| Glyph | LNMax | 4,000 | 4.3 | 72.4% | 82.2% |
| | Confident-GNMax ($T$=1000, $\sigma_1$=500, $\sigma_2$=100) | 10,762 | 2.03 | **75.5%** | |
| | Interactive-GNMax, two rounds | 4,341 | **0.837** | 73.2% | |

Table 1: **Utility and privacy of the students.** The Confident- and Interactive-GNMax aggregators introduced in Section 4 offer better tradeoffs between privacy (characterized by the value of the bound $\varepsilon$) and utility (the accuracy of the student compared to a non-private baseline) than the LNMax aggregator used by the original PATE proposal on all datasets we evaluated with. For MNIST, Adult, and SVHN, we use the labels of ensembles of 250 teachers published by Papernot et al. (2017) and set $\delta = 10^{-5}$ to compute values of $\varepsilon$ (to the exception of SVHN where $\delta = 10^{-6}$). All Glyph results use an ensemble of 5000 teachers and $\varepsilon$ is computed for $\delta = 10^{-8}$.

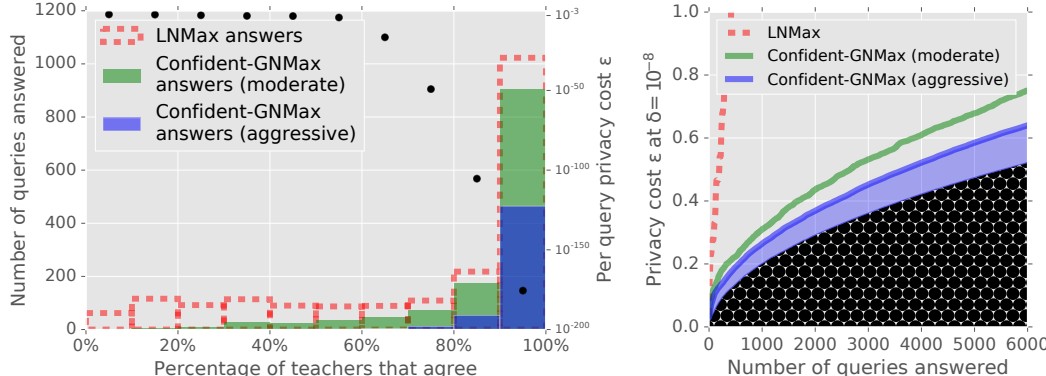

Figure 5: **Effects of the noisy threshold checking:** *Left:* The number of queries answered by LNMax, Confident-GNMax moderate ($T$=3500, $\sigma_1$=1500), and Confident-GNMax aggressive ($T$=5000, $\sigma_1$=1500). The black dots and the right axis (in log scale) show the expected cost of answering a single query in each bin (via GNMax, $\sigma_2$=100). *Right:* Privacy cost of answering all (LNMax) vs only inexpensive queries (GNMax) for a given number of answered queries. The very dark area under the curve is the cost of selecting queries; the rest is the cost of answering them.

Aggregator. A grid search targeting the best privacy for roughly 3,400 answered queries (out of 6,000)—sufficient to bootstrap a student—led us to setting ($T$=3500, $\sigma_1$=1500, $\sigma_2$=100) and a privacy cost of $\varepsilon \approx 0.59$.

In round two, this student was then trained with 10,000 more queries made with the Interactive-GNMax Aggregator ($T$=3500, $\sigma_1$=2000, $\sigma_2$=200). We computed the resulting (total) privacy cost and utility at an *exemplar* data point through another grid search of plausible parameter values. The result appears in the last row of Table 1. With just over 10,422 answered queries in total at a privacy cost of $\varepsilon = 0.84$, the trained student was able to achieve 73.2% accuracy. Note that this students required fewer answered queries compared to the Confident Aggregator. The best overall cost of student training occurred when the privacy costs for the first and second rounds of training were roughly the same. (The total $\varepsilon$ is less than $0.59 \times 2 = 1.18$ due to better composition—via Theorems 4 and 5.)

**Comparison with Baseline.** Note that the Glyph student's accuracy remains seven percentage points below the non-private model's accuracy achieved by training on the 65M sensitive inputs. We hypothesize that this is due to the uncurated nature of the data considered. Indeed, the class imbalance naturally requires more queries to return labels from the less represented classes. For instance, a model trained on 200K queries is only 77% accurate on test data. In addition, the large fraction of mislabeled inputs are likely to have a large privacy cost: these inputs are sensitive because they are outliers of the distribution, which is reflected by the weak consensus among teachers on these inputs.

## 5.4 NOISY THRESHOLD CHECKS AND PRIVACY COSTS

Sections 4.1 and 4.2 motivated the need for a noisy threshold checking step before having the teachers answer queries: it prevents most of the privacy budget being consumed by few queries that are expensive and also likely to be incorrectly answered. In Figure 5, we compare the privacy cost $\varepsilon$ of answering all queries to only answering confident queries for a fixed number of queries.

We run additional experiments to support the evaluation from Section 5.3. With the votes of 5,000 teachers on the Glyph dataset, we plot in Figure 5 the histogram of the plurality vote counts ($n_{i^*}$ in the notation of Section 4.1) across 25,000 student queries. We compare these values to the vote counts of queries that passed the noisy threshold check for two sets of parameters $T$ and $\sigma_1$ in Algorithm 1. Smaller values imply weaker teacher agreements and consequently more expensive queries.

When ($T$=3500, $\sigma_1$=1500) we capture a significant fraction of queries where teachers have a strong consensus (roughly $> 4000$ votes) while managing to filter out many queries with poor consensus. This *moderate check* ensures that although many queries with plurality votes between 2,500 and 3,500 are answered (i.e., only 50–70% of teachers agree on a label) the expensive ones are most likely discarded. For ($T$=5000, $\sigma_1$=1500), queries with poor consensus are completely culled out. This selectivity comes at the expense of a noticeable drop for queries that might have had a strong consensus and little-to-no privacy cost. Thus, this *aggressive check* answer fewer queries with very strong privacy guarantees. We reiterate that this threshold checking step itself is done in a private manner. Empirically, in our Interactive Aggregator experiments, we expend about a third to a half of our privacy budget on this step, which still yields a very small cost *per query* across 6,000 queries.

## 6 Conclusions

The key insight motivating the addition of a noisy thresholding step to the two aggregation mechanisms proposed in our work is that there is a form of synergy between the privacy and accuracy of labels output by the aggregation: *labels that come at a small privacy cost also happen to be more likely to be correct*. As a consequence, we are able to provide more quality supervision to the student by choosing not to output labels when the consensus among teachers is too low to provide an aggregated prediction at a small cost in privacy. This observation was further confirmed in some of our experiments where we observed that if we trained the student on either private or non-private labels, the former almost always gave better performance than the latter—for a fixed number of labels.

Complementary with these aggregation mechanisms is the use of a Gaussian (rather than Laplace) distribution to perturb teacher votes. In our experiments with Glyph data, these changes proved essential to preserve the accuracy of the aggregated labels—because of the large number of classes. The analysis presented in Section 4 details the delicate but necessary adaptation of analogous results for the Laplace NoisyMax.

As was the case for the original PATE proposal, semi-supervised learning was instrumental to ensure the student achieves strong utility given a limited set of labels from the aggregation mechanism. However, we found that virtual adversarial training outperforms the approach from Salimans et al. (2016) in our experiments with Glyph data. These results establish lower bounds on the performance that a student can achieve when supervised with our aggregation mechanisms; future work may continue to investigate virtual adversarial training, semi-supervised generative adversarial networks and other techniques for learning the student in these particular settings with restricted supervision.

## Acknowledgments

We are grateful to Martín Abadi, Vincent Vanhoucke, and Daniel Levy for their useful inputs and discussions towards this paper.

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

# A   APPENDIX: PRIVACY ANALYSIS

In this appendix, we provide the proofs of Theorem 6 and Proposition 7. Moreover, we present Proposition 10, which provides optimal values of $\mu_1$ and $\mu_2$ to apply towards Theorem 6 for the GNMax mechanism. We start off with a statement about the Rényi differential privacy guarantee of the GNMax.

**Proposition 8.** *The GNMax aggregator $\mathcal{M}_\sigma$ guarantees $\left(\lambda, \lambda/\sigma^2\right)$-RDP for all $\lambda \geq 1$.*

*Proof.* The result follows from observing that $\mathcal{M}_\sigma$ can be decomposed into applying the $\mathrm{argmax}$ operator to a noisy histogram resulted from adding Gaussian noise to each dimension of the original histogram. The Gaussian mechanism satisfies $(\lambda, \lambda/2\sigma^2)$-RDP (Mironov, 2017), and since each teacher may change two counts (incrementing one and decrementing the other), the overall RDP guarantee is as claimed. ∎

**Proposition 7.** *For a GNMax aggregator $\mathcal{M}_\sigma$, the teachers' votes histogram $\bar{n} = (n_1, \ldots, n_m)$, and for any $i^* \in [m]$, we have*
$$\mathbf{Pr}\left[\mathcal{M}_\sigma(D) \neq i^*\right] \leq q(\bar{n}),$$
*where*
$$q(\bar{n}) \triangleq \frac{1}{2} \sum_{i \neq i^*} \mathrm{erfc}\left(\frac{n_{i^*} - n_i}{2\sigma}\right).$$

*Proof.* Recall that $\mathcal{M}_\sigma(D) = \mathrm{argmax}(n_i + Z_i)$, where $Z_i$ are distributed as $\mathcal{N}(0, \sigma^2)$. Then for any $i^* \in [m]$, we have

$$\mathbf{Pr}[\mathcal{M}_\sigma(D) \neq i^*] = \mathbf{Pr}\left[\exists i, n_i + Z_i > n_{i^*} + Z_{i^*}\right] \leq \sum_{i \neq i^*} \mathbf{Pr}\left[n_i + Z_i > n_{i^*} + Z_{i^*}\right]$$
$$= \sum_{i \neq i^*} \mathbf{Pr}\left[Z_i - Z_{i^*} > n_{i^*} - n_i\right]$$
$$= \sum_{i \neq i^*} \frac{1}{2}\left(1 - \mathrm{erf}\left(\frac{n_{i^*} - n_i}{2\sigma}\right)\right).$$

where the last equality follows from the fact that $Z_i - Z_j$ is a Gaussian random variable with mean zero and variance $2\sigma^2$. ∎

We now present a precise statement of Theorem 6.

**Theorem 6.** *Let $\mathcal{M}$ be a randomized algorithm with $(\mu_1, \varepsilon_1)$-RDP and $(\mu_2, \varepsilon_2)$-RDP guarantees and suppose that there exists a likely outcome $i^*$ given a dataset $D$ and a bound $\tilde{q} \leq 1$ such that $\tilde{q} \geq \mathbf{Pr}\left[\mathcal{M}(D) \neq i^*\right]$. Additionally suppose that $\lambda \leq \mu_1$ and $\tilde{q} \leq e^{(\mu_2-1)\varepsilon_2}/\left(\frac{\mu_1}{\mu_1-1} \cdot \frac{\mu_2}{\mu_2-1}\right)^{\mu_2}$. Then, for any neighboring dataset $D'$ of $D$, we have:*

$$D_\lambda(\mathcal{M}(D)\|\mathcal{M}(D')) \leq \frac{1}{\lambda - 1} \log\left((1 - \tilde{q}) \cdot \boldsymbol{A}(\tilde{q}, \mu_2, \varepsilon_2)^{\lambda-1} + \tilde{q} \cdot \boldsymbol{B}(\tilde{q}, \mu_1, \varepsilon_1)^{\lambda-1}\right) \quad (2)$$

*where $\boldsymbol{A}(\tilde{q}, \mu_2, \varepsilon_2) \triangleq (1 - \tilde{q})/\left(1 - (\tilde{q}e^{\varepsilon_2})^{\frac{\mu_2-1}{\mu_2}}\right)$ and $\boldsymbol{B}(\tilde{q}, \mu_1, \varepsilon_1) \triangleq e^{\varepsilon_1}/\tilde{q}^{\frac{1}{\mu_1-1}}$.*

*Proof.* Before we proceed to the proof, we introduce some simplifying notation. For a randomized mechanism $\mathcal{M}$ and neighboring datasets $D$ and $D'$, we define

$$\boldsymbol{\beta}_\mathcal{M}(\lambda; D, D') \triangleq D_\lambda(\mathcal{M}(D)\|\mathcal{M}(D'))$$
$$= \frac{1}{\lambda - 1} \log \mathbb{E}_{x \sim \mathcal{M}(D)}\left[\left(\frac{\mathbf{Pr}\left[\mathcal{M}(D) = x\right]}{\mathbf{Pr}\left[\mathcal{M}(D') = x\right]}\right)^{\lambda-1}\right].$$

As the proof involves working with the RDP bounds in the exponent, we set $\zeta_1 \triangleq e^{\varepsilon_1(\mu_1-1)}$ and $\zeta_2 \triangleq e^{\varepsilon_2(\mu_2-1)}$.

Finally, we define the following shortcuts:

$$q_i \triangleq \mathbf{Pr}\left[\mathcal{M}(D) = i\right] \text{ and } q \triangleq \sum_{i \neq i^*} q_i = \mathbf{Pr}\left[\mathcal{M}(D) \neq i^*\right],$$

$$p_i \triangleq \mathbf{Pr}\left[\mathcal{M}(D') = i\right] \text{ and } p \triangleq \sum_{i \neq i^*} p_i = \mathbf{Pr}\left[\mathcal{M}(D') \neq i^*\right],$$

and note that $q \leq \tilde{q}$.

From the definition of Rényi differential privacy, $(\mu_1, \varepsilon_1)$-RDP implies:

$$\exp\left(\boldsymbol{\beta}_{\mathcal{M}}(\mu_1; D, D')\right) = \left(\frac{(1-q)^{\mu_1}}{(1-p)^{\mu_1-1}} + \sum_{i \neq i*} \frac{q_i^{\mu_1}}{p_i^{\mu_1-1}}\right)^{1/(\mu_1-1)} \leq \exp(\varepsilon_1)$$

$$\implies \sum_{i>1} \frac{q_i^{\mu_1}}{p_i^{\mu_1-1}} = \sum_{i>1} q_i \left(\frac{q_i}{p_i}\right)^{\mu_1-1} \leq \zeta_1. \tag{3}$$

Since $\mu_1 \geq \lambda$, $f(x) \triangleq x^{\frac{\mu_1-1}{\lambda-1}}$ is convex. Applying Jensen's Inequality we have the following:

$$\left(\frac{\sum_{i \neq i^*} q_i \left(\frac{q_i}{p_i}\right)^{\lambda-1}}{q}\right)^{\frac{\mu_1-1}{\lambda-1}} \leq \frac{\sum_{i \neq i^*} q_i \left(\frac{q_i}{p_i}\right)^{\mu_1-1}}{q}$$

$$\implies \sum_{i \neq i^*} q_i \left(\frac{q_i}{p_i}\right)^{\lambda-1} \leq q \left(\frac{\sum_{i \neq i^*} q_i \left(\frac{q_i}{p_i}\right)^{\mu_1-1}}{q}\right)^{\frac{\lambda-1}{\mu_1-1}}$$

$$\overset{(3)}{\implies} \sum_{i \neq i^*} q_i \left(\frac{q_i}{p_i}\right)^{\lambda-1} \leq \zeta_1^{\frac{\lambda-1}{\mu_1-1}} \cdot q^{1-\frac{\lambda-1}{\mu_1-1}}. \tag{4}$$

Next, by the bound at order $\mu_2$, we have:

$$\exp\left(\boldsymbol{\beta}_{\mathcal{M}}(\mu_2; D', D)\right) = \left(\frac{(1-p)^{\mu_2}}{(1-q)^{\mu_2-1}} + \sum_{i \neq i^*} \frac{p_i^{\mu_2}}{q_i^{\mu_2-1}}\right)^{1/(\mu_2-1)} \leq \exp(\varepsilon_2)$$

$$\implies \frac{(1-p)^{\mu_2}}{(1-q)^{\mu_2-1}} + \sum_{i \neq i^*} \frac{p_i^{\mu_2}}{q_i^{\mu_2-1}} \leq \zeta_2.$$

By the data processing inequality of Rényi divergence, we have

$$\frac{(1-p)^{\mu_2}}{(1-q)^{\mu_2-1}} + \frac{p^{\mu_2}}{q^{\mu_2-1}} \leq \zeta_2,$$

which implies $\frac{p^{\mu_2}}{q^{\mu_2-1}} \leq \zeta_2$ and thus

$$p \leq \left(q^{\mu_2-1}\zeta_2\right)^{\frac{1}{\mu_2}}. \tag{5}$$

Combining (4) and (5), we can derive a bound at $\lambda$.

$$\exp\left(\boldsymbol{\beta}_{\mathcal{M}}(\lambda, D, D')\right) = \left(\frac{(1-q)^{\lambda}}{(1-p)^{\lambda-1}} + \sum_{i \neq i^*} \frac{q_i^{\lambda}}{p_i^{\lambda-1}}\right)^{1/(\lambda-1)}$$

$$\leq \left(\frac{(1-q)^{\lambda}}{\left(1 - (q^{\mu_2-1}\zeta_2)^{\frac{1}{\mu_2}}\right)^{\lambda-1}} + \zeta_1^{\frac{\lambda-1}{\mu_1-1}} \cdot q^{1-\frac{\lambda-1}{\mu_1-1}}\right)^{1/(\lambda-1)}. \tag{6}$$

Although Equation (6) is very close to the corresponding statement in the theorem's claim, one subtlety remains. The bound (6) applies to the exact probability $q = \mathbf{Pr}\left[\mathcal{M}(D) \neq i^*\right]$. In the theorem statement, and in practice, we can only derive an upper bound $\tilde{q}$ on $\mathbf{Pr}\left[\mathcal{M}(D) \neq i^*\right]$. The last step of the proof requires showing that the expression in Equation (6) is monotone in the range of values of $q$ that we care about.

**Lemma 9** (Monotonicity of the bound). *Let the functions $f_1(\cdot)$ and $f_2(\cdot)$ be*

$$f_1(x) \triangleq \frac{(1-x)^\lambda}{\left(1-(x^{\mu_2-1}\zeta_2)^{\frac{1}{\mu_2}}\right)^{\lambda-1}} \quad and \quad f_2(x) \triangleq \zeta_1^{\frac{\lambda-1}{\mu_1-1}} \cdot x^{1-\frac{\lambda-1}{\mu_1-1}},$$

*Then $f_1(x) + f_2(x)$ is increasing in $\left[0, \min\left(1, \zeta_2/\left(\frac{\mu_1}{\mu_1-1} \cdot \frac{\mu_2}{\mu_2-1}\right)^{\mu_2}\right)\right]$.*

*Proof.* Taking the derivative of $f_1(x)$, we have:

$$f_1'(x) = \frac{-\lambda(1-x)^{\lambda-1}(1-(x^{\mu_2-1}\zeta_2)^{\frac{1}{\mu_2}})^{\lambda-1}}{(1-(x^{\mu_2-1}\zeta_2)^{\frac{1}{\mu_2}})^{2\lambda-2}}$$

$$+ \frac{(1-x)^\lambda(\lambda-1)(1-(x^{\mu_2-1}\zeta_2)^{\frac{1}{\mu_2}})^{\lambda-2}\zeta_2^{\frac{1}{\mu_2}} \cdot \frac{\mu_2-1}{\mu_2} \cdot x^{-\frac{1}{\mu_2}}}{(1-(x^{\mu_2-1}\zeta_2)^{\frac{1}{\mu_2}})^{2\lambda-2}}$$

$$= \frac{(1-x)^{\lambda-1}}{(1-(x^{\mu_2-1}\zeta_2)^{\frac{1}{\mu_2}})^{\lambda-1}}\left(-\lambda + (\lambda-1)\left(1-\frac{1}{\mu_2}\right)\frac{1-x}{1-(x^{\mu_2-1}\zeta_2)^{\frac{1}{\mu_2}}}\left(\frac{\zeta_2}{x}\right)^{\frac{1}{\mu_2}}\right).$$

We intend to show that:

$$f_1'(x) \geq -\lambda + (\lambda-1)\left(1-\frac{1}{\mu_2}\right)\left(\frac{\zeta_2}{x}\right)^{\frac{1}{\mu_2}}. \tag{7}$$

For $x \in \left[0, \zeta_2/\left(\frac{\mu_1}{\mu_1-1} \cdot \frac{\mu_2}{\mu_2-1}\right)^{\mu_2}\right]$ and $y \in [1, \infty)$, define $g(x, y)$ as:

$$g(x, y) \triangleq -\lambda \cdot y^{\lambda-1} + (\lambda-1)\left(1-\frac{1}{\mu_2}\right)\left(\frac{\zeta_2}{x}\right)^{\frac{1}{\mu_2}}y^\lambda.$$

We claim that $g(x, y)$ is increasing in $y$ and therefore $g(x, y) \geq g(x, 1)$, and prove it by showing the partial derivative of $g(x, y)$ with respect to $y$ is non-negative. Take a derivative with respect to $y$ as:

$$g_y'(x, y) = -\lambda(\lambda-1)y^{\lambda-2} + \lambda(\lambda-1)\left(1-\frac{1}{\mu_2}\right)\left(\frac{\zeta_2}{x}\right)^{\frac{1}{\mu_2}}y^{\lambda-1}$$

$$= \lambda(\lambda-1)y^{\lambda-2}\left(-1 + \left(1-\frac{1}{\mu_2}\right)\left(\frac{\zeta_2}{x}\right)^{\frac{1}{\mu_2}}y\right).$$

To see why $g_y'(x, y)$ is non-negative in the respective ranges of $x$ and $y$, note that:

$$x \leq \zeta_2/\left(\frac{\mu_1}{\mu_1-1} \cdot \frac{\mu_2}{\mu_2-1}\right)^{\mu_2} \implies x \leq \zeta_2/\left(\frac{\mu_2}{\mu_2-1}\right)^{\mu_2}$$

$$\implies 1 \leq \frac{\zeta_2}{x} \cdot \left(\frac{\mu_2-1}{\mu_2}\right)^{\mu_2}$$

$$\implies 1 \leq \frac{\mu_2-1}{\mu_2}\left(\frac{\zeta_2}{x}\right)^{\frac{1}{\mu_2}}$$

$$\implies 1 \leq \frac{\mu_2-1}{\mu_2}\left(\frac{\zeta_2}{x}\right)^{\frac{1}{\mu_2}}y \qquad \text{(as } y \geq 1\text{)}$$

$$\implies 0 \leq -1 + \frac{\mu_2-1}{\mu_2}\left(\frac{\zeta_2}{x}\right)^{\frac{1}{\mu_2}}y$$

$$\implies 0 \leq g_y'(x, y). \qquad \text{(in the resp. range of } x \text{ and } y\text{)}$$

Consider $\frac{1-x}{1-(x^{\mu_2-1}\zeta_2)^{1/\mu_2}}$. Since $\zeta_2 \geq 1$ and $x \leq 1$, we have $x \leq \zeta_2$ and hence

$$\frac{1-x}{1-(x^{\mu_2-1}\zeta_2)^{\frac{1}{\mu_2}}} \geq \frac{1-x}{1-(x^{\mu_2-1}x)^{\frac{1}{\mu_2}}} = 1.$$

Therefore we can set $y = \frac{1-x}{1-(x^{\mu_2-1}\zeta_2)^{1/\mu_2}}$ and apply the fact that $g(x,y) \geq g(x,1)$ for all $y \geq 1$ to get

$$f_1'(x) \geq -\lambda + (\lambda - 1)\left(1 - \frac{1}{\mu_2}\right)\left(\frac{\zeta_2}{x}\right)^{\frac{1}{\mu_2}},$$

as required by (7).

Taking the derivative of $f_2(x)$, we have:

$$f_2'(x) = \zeta_1^{\frac{\lambda-1}{\mu_1-1}} \cdot \left(1 - \frac{\lambda-1}{\mu_1-1}\right) x^{-\frac{\lambda-1}{\mu_1-1}} = \left(\frac{\zeta_1}{x}\right)^{\frac{\lambda-1}{\mu_1-1}}\left(1 - \frac{\lambda-1}{\mu_1-1}\right) \geq 1 - \frac{\lambda-1}{\mu_1-1}.$$

Combining the two terms together, we have:

$$f'(x) \geq -\lambda + (\lambda-1)\left(1 - \frac{1}{\mu_2}\right)\left(\frac{\zeta_2}{x}\right)^{\frac{1}{\mu_2}} + 1 - \frac{\lambda-1}{\mu_1-1}$$

$$= (\lambda-1)\left(-\frac{\mu_1}{\mu_1-1} + \frac{\mu_2-1}{\mu_2}\left(\frac{\zeta_2}{x}\right)^{\frac{1}{\mu_2}}\right).$$

For $f'(x)$ to be non-negative we need:

$$-\frac{\mu_1}{\mu_1-1} + \frac{\mu_2-1}{\mu_2}\left(\frac{\zeta_2}{x}\right)^{\frac{1}{\mu_2}} \geq 0$$

$$\iff \left(\frac{\mu_1}{\mu_1-1} \cdot \frac{\mu_2}{\mu_2-1}\right)^{\mu_2} \leq \frac{\zeta_2}{x}.$$

So $f(x)$ is increasing for $x \in \left[0, \zeta_2/\left(\frac{\mu_1}{\mu_1-1} \cdot \frac{\mu_2}{\mu_2-1}\right)^{\mu_2}\right]$. This means for $q \leq \tilde{q} \leq \zeta_2/\left(\frac{\mu_1}{\mu_1-1} \cdot \frac{\mu_2}{\mu_2-1}\right)^{\mu_2}$, we have $f(q) \leq f(\tilde{q})$. This completes the proof of the lemma and that of the theorem. ∎

∎

Theorem 6 yields data-dependent Rényi differential privacy bounds for *any* value of $\mu_1$ and $\mu_2$ larger than $\lambda$. The following proposition simplifies this search by calculating optimal higher moments $\mu_1$ and $\mu_2$ for the GNMax mechanism with variance $\sigma^2$.

**Proposition 10.** *When applying Theorem 6 and Proposition 8 for GNMax with Gaussian of variance $\sigma^2$, the right-hand side of (2) is minimized at*

$$\mu_2 = \sigma \cdot \sqrt{\log(1/\tilde{q})}, \text{ and } \mu_1 = \mu_2 + 1.$$

*Proof.* We can minimize both terms in (2) independently. To minimize the first term in (6), we minimize $(\tilde{q}e^{\varepsilon_2})^{1-1/\mu_2}$ by considering logarithms:

$$\log\left\{(\tilde{q}e^{\varepsilon_2})^{1-1/\mu_2}\right\} = \log\left\{\tilde{q}^{1-\frac{1}{\mu_2}}\exp\left(\frac{\mu_2-1}{\sigma^2}\right)\right\}$$

$$= \left(1 - \frac{1}{\mu_2}\right) \cdot \log\tilde{q} + \frac{\mu_2-1}{\sigma^2}$$

$$= \frac{1}{\mu_2}\log\frac{1}{\tilde{q}} + \frac{\mu_2}{\sigma^2} - \frac{1}{\sigma^2} - \log\frac{1}{\tilde{q}},$$

which is minimized at $\mu_2 = \sigma \cdot \sqrt{\log(1/\tilde{q})}$.

To minimize the second term in (6), we minimize $e^{\varepsilon_1}/\tilde{q}^{1/(\mu_1-1)}$ as follows:

$$
\begin{aligned}
\log\left\{\frac{e^{\varepsilon_1}}{\tilde{q}^{1/(\mu_1-1)}}\right\} &= \log\left\{\tilde{q}^{-1/(\mu_1-1)}\exp\left(\frac{\mu_1}{\sigma^2}\right)\right\} \\
&= \frac{\mu_1}{\sigma^2} + \frac{1}{\mu_1-1}\log\frac{1}{\tilde{q}} \\
&= \frac{1}{\sigma^2} + \frac{\mu_1-1}{\sigma^2} + \frac{1}{\mu_1-1}\log\frac{1}{\tilde{q}},
\end{aligned}
$$

which is minimized at $\mu_1 = 1 + \sigma \cdot \sqrt{\log(1/\tilde{q})}$ completing the proof. ∎

Putting this together, we apply the following steps to calculate RDP of order $\lambda$ for GNMax with variance $\sigma^2$ on a given dataset $D$. First, we compute a bound $q$ according to Proposition 7. Then we use the smaller of two bounds: a data-dependent (Theorem 6) and a data-independent one (Proposition 8) :

$$
\boldsymbol{\beta}_\sigma(q) \triangleq \min\left\{\frac{1}{\lambda-1}\log\left\{(1-q)\cdot\boldsymbol{A}(q,\mu_2,\varepsilon_2)^{\lambda-1} + q\cdot\boldsymbol{B}(q,\mu_1,\varepsilon_1)^{\lambda-1}\right\}, \lambda/\sigma^2\right\},
$$

where $\boldsymbol{A}$ and $\boldsymbol{B}$ are defined as in the statement of Theorem 6, the parameters $\mu_1$ and $\mu_2$ are selected according to Proposition 10, and $\varepsilon_1 \triangleq \mu_1/\sigma^2$ and $\varepsilon_2 \triangleq \mu_2/\sigma^2$ (Proposition 8). Importantly, the first expression is evaluated only when $q < 1$, $\mu_1 \geq \lambda$, $\mu_2 > 1$, and $q \leq e^{(\mu_2-1)\varepsilon_2}/\left(\frac{\mu_1}{\mu_1-1}\cdot\frac{\mu_2}{\mu_2-1}\right)^{\mu_2}$. These conditions can either be checked for each application of the aggregation mechanism, or a critical value of $q_0$ that separates the range of applicability of the data-dependent and data-independent bounds can be computed for given $\sigma$ and $\lambda$. In our implementation we pursue the second approach.

The following corollary offers a simple asymptotic expression of the privacy of GNMax for the case when there are large (relative to $\sigma$) gaps between the highest *three* vote counts.

**Corollary 11.** *If the top three vote counts are $n_1 > n_2 > n_3$ and $n_1 - n_2, n_2 - n_3 \gg \sigma$, then the mechanism GNMax with Gaussian of variance $\sigma^2$ satisfies $(\lambda, \exp(-2\lambda/\sigma^2)/\lambda)$-RDP for $\lambda = (n_1 - n_2)/4$.*

*Proof.* Denote the noisy counts as $\tilde{n}_i = n_i + \mathcal{N}(0, \sigma^2)$. Ignoring outputs other than those with the highest and the second highest counts, we bound $q = \mathbf{Pr}\left[\mathcal{M}(D) \neq 1\right]$ as $\mathbf{Pr}[\tilde{n}_1 < \tilde{n}_2] = \mathbf{Pr}[N(0, 2\sigma^2) > n_1 - n_2] < \exp\left(-(n_1-n_2)^2/4\sigma^2\right)$, which we use as $\tilde{q}$. Plugging $\tilde{q}$ in Proposition 10, we have $\mu_1 - 1 = \mu_2 = (n_1 - n_2)/2$, limiting the range of applicability of Theorem 6 to $\lambda < (n_1 - n_2)/2$.

Choosing $\lambda = (n_1 - n_2)/4$ ensures $\boldsymbol{A}(\tilde{q}, \mu_2, \varepsilon_2) \approx 1$, which allows approximating the bound (2) as $\tilde{q}\cdot\boldsymbol{B}(\tilde{q},\mu_1,\varepsilon_1)^{\lambda-1}/(\lambda-1)$. The proof follows by straightforward calculation. ∎

## B  SMOOTH SENSITIVITY AND PUBLISHING THE PRIVACY PARAMETER

The privacy guarantees obtained for the mechanisms in this paper via Theorem 6 take as input $\tilde{q}$, an upper bound on the probability that the aggregate mechanism returns the true plurality. This means that the resulting privacy parameters computed depend on teacher votes and hence the underlying data. To avoid potential privacy breaches from simply publishing the data-dependent parameter, we need to publish a *sanitized version* of the privacy loss. This is done by adding noise to the computed privacy loss estimates using the *smooth sensitivity* algorithm proposed by Nissim et al. (2007).

This section has the following structure. First we recall the notion of smooth sensitivity and introduce an algorithm for computing the smooth sensitivity of the privacy loss function of the GNMax mechanism. In the rest of the section we prove correctness of these algorithms by stating several conditions on the mechanism, proving that these conditions are sufficient for correctness of the algorithm, and finally demonstrating that GNMax satisfies these conditions.

### B.1 COMPUTING SMOOTH SENSITIVITY

Any dataset $D$ defines a histogram $\bar{n} = (n_1, \ldots, n_m) \in \mathbb{N}^m$ of the teachers' votes. We have a natural notion of the distance between two histograms $\text{dist}(\bar{n}, \bar{n}')$ and a function $q: \mathbb{N}^m \to [0, 1]$ on these histograms computing the bound according to Proposition 7. The value $q(\bar{n})$ can be used as $\tilde{q}$ in the application of Theorem 6. Additionally we have $n^{(i)}$ denote the $i$-th highest bar in the histogram.

We aim at calculating a smooth sensitivity of $\boldsymbol{\beta}\left(q(\bar{n})\right)$ whose definition we recall now.

**Definition 12** (Smooth Sensitivity). *Given the smoothness parameter $\beta$, a $\beta$-smooth sensitivity of $f(n)$ is defined as*

$$\text{SS}_\beta(\bar{n}) \triangleq \max_{d \geq 0} e^{-\beta d} \cdot \max_{\bar{n}': \text{dist}(\bar{n}, \bar{n}') \leq d} \tilde{\text{LS}}(\bar{n}'),$$

*where*

$$\tilde{\text{LS}}(\bar{n}) \geq \max_{\bar{n}': \text{dist}(\bar{n}, \bar{n}') = 1} |f(n) - f(n')|$$

*is an upper bound on the local sensitivity.*

We now describe Algorithms 3–5 computing a smooth sensitivity of $\boldsymbol{\beta}\left(q(\cdot)\right)$. The algorithms assume the existence of efficiently computable functions $q: \mathbb{N}^m \to [0, 1]$, $\text{B}_\text{L}, \text{B}_\text{U}: [0, 1] \to [0, 1]$, and a constant $q_0$.

Informally, the functions $\text{B}_\text{U}$ and $\text{B}_\text{L}$ respectively upper and lower bound the value of $q$ evaluated at any neighbor of $\bar{n}$ given $q(\bar{n})$, and $[0, q_0)$ limits the range of applicability of data-dependent analysis.

The functions $\text{B}_\text{L}$ and $\text{B}_\text{U}$ are defined as follows. Their derivation appears in Section B.4.

$$\text{B}_\text{U}(q) \triangleq \min\left\{ \frac{m-1}{2} \operatorname{erfc}\left( \operatorname{erfc}^{\text{-1}}\left( \frac{2q}{m-1} \right) - \frac{1}{\sigma} \right), 1 \right\},$$

$$\text{B}_\text{L}(q) \triangleq \frac{m-1}{2} \operatorname{erfc}\left( \operatorname{erfc}^{\text{-1}}\left( \frac{2q}{m-1} \right) + \frac{1}{\sigma} \right),$$

---

**Algorithm 3 – Local Sensitivity:** use the functions $\text{B}_\text{U}$ and $\text{B}_\text{L}$ to compute (an upper bound) of the local sensitivity at a given $q$ value by looking at the difference of $\boldsymbol{\beta}\left(\cdot\right)$ evaluated on the bounds.

---

1: **procedure** $\tilde{\text{LS}}(q)$
2:     **if** $q_1 \leq q \leq q_0$ **then**                          $\triangleright q_1 = \text{B}_\text{L}(q_0)$. Interpolate the middle part.
3:         $q \leftarrow q_1$
4:     **end if**
5:     **return** $\max\{\boldsymbol{\beta}\left(\text{B}_\text{U}(q)\right) - \boldsymbol{\beta}\left(q\right), \boldsymbol{\beta}\left(q\right) - \boldsymbol{\beta}\left(\text{B}_\text{L}(q)\right)\}$
6: **end procedure**

---

### B.2 NOTATION AND CONDITIONS

**Notation.** We find that the algorithm and the proof of its correctness are more naturally expressed if we relax the notions of a histogram and its neighbors to allow non-integer values.

- We generalize histograms to be any vector with non-negative real values. This relaxation is used only in the analysis of algorithms; the actual computations are performed exclusively over integer-valued inputs.

- Let $\bar{n} = [n_1, \ldots, n_m] \in \mathbb{R}^m$, $n_i \geq 0$ denote a histogram. Let $n^{(i)}$ denote the $i$-th bar in the descending order.

- Define a "move" as increasing one bar by some value in $[0, 1]$ and decreasing one bar by a (possibly different) value in $[0, 1]$ subject to the resulting value be non-negative. Notice the difference between the original problem and our relaxation. In the original formulation, the histogram takes only integer values and we can only increase/decrease them by exactly 1. In contrast, we allow real values and a teacher can contribute an arbitrary amount in $[0, 1]$ to any one class.

---

**Algorithm 4 – Sensitivity at a distance:** given a histogram $\bar{n}$, compute the sensitivity of $\boldsymbol{\beta}(\cdot)$ at distance at most $d$ using the procedure $\tilde{\mathrm{LS}}$, function $q(\cdot)$, constants $q_0$ and $q_1 = \mathrm{B_L}(q_0)$, and careful case analysis that finds the neighbor at distance $d$ with the maximum sensitivity.

---

1: **procedure** ATDISTANCED($\bar{n}, d$)
2:     $q \leftarrow q(\bar{n})$
3:     **if** $q_1 \leq q \leq q_0$ **then**                                                   $\triangleright$ $q$ is in the flat region.
4:         **return** $\tilde{\mathrm{LS}}(q)$, STOP
5:     **end if**
6:     **if** $q < q_1$ **then**                                                              $\triangleright$ Need to increase $q$.
7:         **if** $n^{(1)} - n^{(2)} < 2d$ **then**                                          $\triangleright$ $n^{(i)}$ is the $i$th largest element.
8:             **return** $\tilde{\mathrm{LS}}(q_1)$, STOP
9:         **else**
10:             $\bar{n}' \leftarrow \mathrm{SORT}(\bar{n}) + [-d, d, 0, \ldots, 0]$
11:             $q' \leftarrow q(\bar{n}')$
12:             **if** $q' > q_1$ **then**
13:                 **return** $\tilde{\mathrm{LS}}(q_0)$, STOP
14:             **else**
15:                 **return** $\tilde{\mathrm{LS}}(q')$, CONTINUE
16:             **end if**
17:         **end if**
18:     **else**                                                                          $\triangleright$ Need to decrease $q$.
19:         **if** $\sum_{i=2}^{d} n^{(i)} \leq d$ **then**
20:             $\bar{n}' \leftarrow [n, 0, \ldots, 0]$
21:             $q' \leftarrow q(\bar{n}')$
22:             **return** $\tilde{\mathrm{LS}}(q')$, STOP
23:         **else**
24:             $\bar{n}' \leftarrow \mathrm{SORT}(\bar{n}) + [d, 0, \ldots, 0]$
25:             **for** $d' = 1, \ldots, d$ **do**
26:                 $n'^{(2)} \leftarrow n'^{(2)} - 1$                                       $\triangleright$ The index of $n'^{(2)}$ may change.
27:             **end for**
28:             $q' \leftarrow q(\bar{n}')$
29:             **if** $q' < q_0$ **then**
30:                 **return** $\tilde{\mathrm{LS}}(q_0)$, STOP
31:             **else**
32:                 **return** $\tilde{\mathrm{LS}}(q')$, CONTINUE
33:             **end if**
34:         **end if**
35:     **end if**
36: **end procedure**

---

**Algorithm 5 – Smooth Sensitivity:** Compute the $\beta$ smooth sensitivity of $\boldsymbol{\beta}(\cdot)$ via Definition 12 by looking at sensitivities at various distances and returning the maximum weighted by $e^{-\beta d}$.

---

1: **procedure** SMOOTHSENSITIVITY($\bar{n}, \beta$)
2:     $S \leftarrow 0$
3:     $d \leftarrow 0$
4:     **repeat**
5:         $c, \mathrm{StoppingCondition} \leftarrow \mathrm{ATDISTANCED}(\bar{n}, d)$
6:         $S \leftarrow \max\{S, c \cdot e^{-\beta d}\}$
7:         $d \leftarrow d + 1$
8:     **until** StoppingCondition $=$ STOP
9: **end procedure**

---

• Define the distance between two histograms $\bar{n} = (n_1, \ldots, n_m)$ and $\bar{n}' = (n'_1, \ldots, n'_m)$ as

$$d(\bar{n}, \bar{n}') \triangleq \max\left\{ \sum_{i:n_i>n'_i} \lceil n_i - n'_i \rceil, \quad \sum_{i:n_i<n'_i} \lceil n'_i - n_i \rceil \right\},$$

which is equal to the smallest number of "moves" needed to make the two histograms identical. We use the ceiling function since a single step can increase/decrease one bar by at most 1.

We say that two histograms are *neighbors* if their distance $d$ is 1.

Notice that analyses of Rényi differential privacy for LNMax, GNMax and the exponential mechanism are still applicable when the neighboring datasets are defined in this manner.

- Given a randomized aggregator $\mathcal{M}\colon \mathbb{R}^m_{\geq 0} \to [m]$, let $q\colon \mathbb{R}^m_{\geq 0} \to [0,1]$ be so that

$$q(\bar{n}) \geq \mathbf{Pr}[\mathcal{M}(\bar{n}) \neq \operatorname{argmax}(\bar{n})].$$

  When the context is clear, we use $q$ to denote a specific value of the function, which, in particular, can be used as $\tilde{q}$ in applications of Theorem 6.

- Let $\boldsymbol{\beta}\colon [0,1] \to \mathbb{R}$ be the function that maps a $q$ value to the value of the Rényi accountant.

**Conditions.** Throughout this section we will be referring to the list of conditions on $q(\cdot)$ and $\boldsymbol{\beta}(\cdot)$:

C1. The function $q(\cdot)$ is continuous in each argument $n_i$.

C2. There exist functions $\mathrm{B_U}, \mathrm{B_L}\colon [0,1] \to [0,1]$ such that for any neighbor $\bar{n}'$ of $\bar{n}$, we have $\mathrm{B_L}(q(\bar{n})) \leq q(\bar{n}') \leq \mathrm{B_U}(q(\bar{n}))$, i.e., $\mathrm{B_U}$ and $\mathrm{B_L}$ provide upper and lower bounds on the $q$ value of any neighbor of $\bar{n}$.

C3. $\mathrm{B_L}(q)$ is increasing in $q$.

C4. $\mathrm{B_U}$ and $\mathrm{B_L}$ are functional inverses of each other in part of the range, i.e., $q = \mathrm{B_L}(\mathrm{B_U}(q))$ for all $q \in [0, q_0]$, where $q_0$ is defined below. Additionally $\mathrm{B_L}(q) \leq q \leq \mathrm{B_U}(q)$ for all $q \in [0,1]$.

C5. $\boldsymbol{\beta}(\cdot)$ has the following shape: there exist constants $\beta^*$ and $q_0 \leq 0.5$, such that $\boldsymbol{\beta}(q)$ non-decreasing in $[0, q_0]$ and $\boldsymbol{\beta}(q) = \beta^* \geq \boldsymbol{\beta}(q_0)$ for $q > q_0$. The constant $\beta^*$ corresponds to a data-independent bound.

C6. $\Delta\boldsymbol{\beta}(q) \triangleq \boldsymbol{\beta}(\mathrm{B_U}(q)) - \boldsymbol{\beta}(q)$ is non-decreasing in $[0, \mathrm{B_L}(q_0)]$, i.e., when $\mathrm{B_U}(q) \leq q_0$.

C7. Recall that $n^{(i)}$ is the $i$-th largest coordinate of a histogram $\bar{n}$. Then, if $q(\bar{n}) \leq \mathrm{B_U}(q_0)$, then $q(\bar{n})$ is differentiable in all coordinates and

$$\forall i > j \geq 2 \quad \frac{\partial q}{\partial n^{(j)}}(\bar{n}) \geq \frac{\partial q}{\partial n^{(i)}}(\bar{n}) \geq 0.$$

C8. The function $q(\bar{n})$ is invariant under addition of a constant, i.e.,

$$q(\bar{n}) = q(\bar{n} + [x, \ldots, x]) \text{ for all } \bar{n} \text{ and } x \geq 0,$$

and $q(\bar{n})$ is invariant under permutation of $\bar{n}$, i.e.,

$$q(\bar{n}) = q(\pi(\bar{n})) \text{ for all permutations } \pi \text{ on } [m].$$

Finally, we require that if $n^{(1)} = n^{(2)}$, then $q(\bar{n}) \geq q_0$.

We may additionally assume that $q_0 \geq q([n, 0, \ldots, 0])$. Indeed, if this condition is not satisfied, then the data-dependent analysis is not going to be used anywhere. The most extreme histogram—$[n, 0, \ldots, 0]$—is the most advantageous setting for applying data-dependent bounds. If we cannot use the data-dependent bound even in that case, we would be using the data-independent bound everywhere and do not need to compute smooth sensitivity anyway. Yet this condition is not automatically satisfied. For example, if $m$ (the number of classes) is large compared to $n$ (the number of teachers), we might have large $q([n, 0, \ldots, 0])$. So we need to check this condition in the code before doing smooth sensitivity calculation.

### B.3 CORRECTNESS OF ALGORITHMS 3–5

Recall that local sensitivity of a deterministic function $f$ is defined as $\max f(D) - f(D')$, where $D$ and $D'$ are neighbors.

**Proposition 13.** *Under conditions C2–C6, Algorithm 3 computes an upper bound on local sensitivity of $\boldsymbol{\beta}(q(\bar{n}))$.*

*Proof.* Since $\boldsymbol{\beta}(\cdot)$ is non-decreasing everywhere (by C5), and for any neighbors $\bar{n}$ and $\bar{n}'$ it holds that $\mathrm{B_L}(q(\bar{n})) \leq q(\bar{n}') \leq \mathrm{B_U}(q(\bar{n}))$ (by C2), we have the following

$$|\boldsymbol{\beta}(q(\bar{n})) - \boldsymbol{\beta}(q(\bar{n}'))| \leq \max\left\{\boldsymbol{\beta}\Big(\mathrm{B_U}(q(\bar{n}))\Big) - \boldsymbol{\beta}\Big(q(\bar{n})\Big),\ \boldsymbol{\beta}\Big(q(\bar{n})\Big) - \boldsymbol{\beta}\Big(\mathrm{B_L}(q(\bar{n}))\Big)\right\}$$

$$= \max\left\{\Delta\boldsymbol{\beta}\Big(q(\bar{n})\Big),\ \Delta\boldsymbol{\beta}\Big(\mathrm{B_L}(q(\bar{n}))\Big)\right\}$$

as an upper bound on the local sensitivity of $\boldsymbol{\beta}(q(\cdot))$ at input $\bar{n}$.

The function computed by Algorithm 3 differs from above when $q(\bar{n}) \in (\mathrm{B_L}(q_0), q_0)$. To complete the proof we need to argue that the local sensitivity is upper bounded by $\Delta\boldsymbol{\beta}(\mathrm{B_L}(q_0))$ for $q(\bar{n})$ in this interval. The bound follows from the following three observations.

First, $\Delta\boldsymbol{\beta}(q)$ is non-increasing in the range $(\mathrm{B_L}(q_0), 1]$, since $\boldsymbol{\beta}(\mathrm{B_U}(q))$ is constant (by $\mathrm{B_U}(q) \geq \mathrm{B_U}(\mathrm{B_L}(q_0)) = q_0$ and C5) and $\boldsymbol{\beta}(q)$ is non-decreasing in the range (by C5). In particular,

$$\Delta\boldsymbol{\beta}(q) \leq \Delta\boldsymbol{\beta}(\mathrm{B_L}(q_0)) \text{ if } q \geq \mathrm{B_L}(q_0). \tag{8}$$

Second, $\Delta\boldsymbol{\beta}(\mathrm{B_L}(q))$ is non-decreasing in the range $[0, q_0]$ since $\mathrm{B_L}(q)$ is increasing (by C3 and C6). This implies that

$$\Delta\boldsymbol{\beta}(\mathrm{B_L}(q)) \leq \Delta\boldsymbol{\beta}(\mathrm{B_L}(q_0)) \text{ if } q \leq q_0. \tag{9}$$

By (8) and (9) applied to the intersection of the two ranges, it holds that

$$\max\left\{\Delta\boldsymbol{\beta}\Big(q(\bar{n})\Big),\ \Delta\boldsymbol{\beta}\Big(\mathrm{B_L}(q(\bar{n}))\Big)\right\} \leq \Delta\boldsymbol{\beta}(\mathrm{B_L}(q_0)) \text{ if } \mathrm{B_L}(q_0) \leq q \leq q_0,$$

as needed. ∎

We thus established that the function computed by Algorithm 3, which we call $\tilde{\mathrm{LS}}(q)$ from now on, is an upper bound on the local sensitivity. Formally,

$$\tilde{\mathrm{LS}}(q) \triangleq \begin{cases} \Delta\boldsymbol{\beta}(\mathrm{B_L}(q_0)) & \text{if } q \in (\mathrm{B_L}(q_0), q_0), \\ \max\{\Delta\boldsymbol{\beta}(q), \Delta\boldsymbol{\beta}(\mathrm{B_L}(q))\} & \text{otherwise.} \end{cases}$$

The following proposition characterizes the growth of $\tilde{\mathrm{LS}}(q)$.

**Proposition 14.** *Assuming conditions C2–C6, the function $\tilde{\mathrm{LS}}(q)$ is non-decreasing in $[0, \mathrm{B_L}(q_0)]$, constant in $[\mathrm{B_L}(q_0), q_0]$, and non-increasing in $[q_0, 1]$.*

*Proof.* Consider separately three intervals.

- By construction, $\tilde{\mathrm{LS}}$ is constant in $[\mathrm{B_L}(q_0), q_0]$.

- Since both functions $\Delta\boldsymbol{\beta}(\cdot)$ and $\Delta\boldsymbol{\beta}(\mathrm{B_L}(\cdot))$ are each non-decreasing in $[0, \mathrm{B_L}(q_0))$, so is their max.

- In the interval $(q_0, 1]$, $\boldsymbol{\beta}(q)$ is constant. Hence $\Delta\boldsymbol{\beta}(q) = 0$ and $\Delta\boldsymbol{\beta}(\mathrm{B_L}(q)) = \boldsymbol{\beta}(q) - \boldsymbol{\beta}(\mathrm{B_L}(q))$ is non-decreasing. Their maximum value $\Delta\boldsymbol{\beta}(\mathrm{B_L}(q))$ is non-decreasing.

The claim follows. ∎

We next prove correctness of Algorithm 4, which computes the maximal sensitivity of $\boldsymbol{\beta}$ at a fixed distance.

The proof relies on the following notion of a partial order between histograms.

**Definition 15.** *Prefix sums $S_i(\bar{n})$ are defined as follows:*

$$S_i(\bar{n}) \triangleq \sum_{j=1}^{i} (n^{(1)} - n^{(j)}).$$

*We say that a histogram $\bar{n}$ dominates $\bar{n}'$, denoted as $\bar{n} \succeq \bar{n}'$, iff:*

$$\forall (1 < i \leq m) \text{ it holds that } S_i(\bar{n}) \geq S_i(\bar{n}').$$

The function $q(\cdot)$ is monotone under this notion of dominance (assuming certain conditions hold):

**Proposition 16.** *If $q(\cdot)$ satisfies C1, C2, C7, and C8, and $q(\bar{n}) < \mathrm{B_U}(q_0)$, then*

$$\bar{n} \succeq \bar{n}' \Rightarrow q(\bar{n}) \leq q(\bar{n}').$$

*Proof.* We may assume that $n^{(1)} = n'^{(1)}$. Indeed, if this does not hold, add $|n^{(1)} - n'^{(1)}|$ to all coordinates of the histogram with the smaller of the two values. This transform does not change the $q$ value (by C8) and it preserves the $\succeq$ relationship as all prefix sums $S_i(\cdot)$ remain unchanged.

We make a simple observation that will be helpful later:

$$\forall i \in [m] \text{ it holds that } \sum_{j=1}^{i} (n^{(1)} - n_j) \geq S_i(\bar{n}). \tag{10}$$

The inequality holds because the prefix sum accumulates the gaps between the largest value of $\bar{n}$ and all other values in the non-decreasing order. Any deviation from this order may only increase the prefix sums.

The following lemma constructs a monotone chain (in the partial order of dominance) of histograms connecting $\bar{n}$ and $\bar{n}'$ via a sequence of intermediate steps that either do not change the value of $q$ or touch at most two coordinates at a time.

**Lemma 17.** *There exists a chain $\bar{n} = \bar{n}_0 \succeq \bar{n}_1 \succeq \cdots \succeq \bar{n}_d = \bar{n}'$, such that for all $i \in [d]$ either $d(\bar{n}_{i-1}, \bar{n}_i) = 1$ or $\bar{n}_{i-1} = \pi(\bar{n}_i)$ for some permutation $\pi$ on $[m]$. Additionally, $n_0^{(1)} = \cdots = n_d^{(1)}$.*

*Proof (Lemma).* Wlog we assume that $\bar{n}$ and $\bar{n}'$ are each sorted in the descending order. The proof is by induction on $\ell(\bar{n}, \bar{n}') \triangleq \sum_i \lceil n_i - n_i' \rceil \leq 2d(\bar{n}, \bar{n}')$, which, by construction, only assumes non-negative integer values.

If the distance is 0, the statement is immediate. Otherwise, find the smallest $i$ so that $S_i(\bar{n}) > S_i(\bar{n}')$ (if all prefix sums are equal and $n^{(1)} = n'^{(1)}$, it would imply that $\bar{n} = \bar{n}'$). In particular, it means that $n_j = n_j'$ for $j < i$ and $n_i < n_i' \leq n_{i-1} = n_{i-1}'$. Let $x \triangleq \min(n_i' - n_i, 1)$. Define $\bar{n}''$ as identical to $\bar{n}'$ except that $n_i'' = n_i' - x$. The new value is guaranteed to be non-negative, since $x \leq n_i' - n_i$ and $n_i \geq 0$. Note that $\bar{n}''$ is not necessarily sorted anymore. Consider two possibilities.

**Case I:** $\bar{n} \succeq \bar{n}''$. Since $\bar{n}'' \succeq \bar{n}'$, $\ell(\bar{n}, \bar{n}'') < \ell(\bar{n}, \bar{n}')$, and $d(\bar{n}'', \bar{n}') = 1$, we may apply the induction hypothesis to the pair $\bar{n}, \bar{n}''$.

**Case II:** $\bar{n} \not\succeq \bar{n}''$. This may happen because the prefix sums of $\bar{n}''$ increase compared to $S_j(\bar{n}')$ for $j \geq i$. Find the smallest such $i'$ so that $\sum_{j=1}^{i'} (n_1'' - n_j'') > S_{i'}(\bar{n})$. (Since $\bar{n}''$ is not sorted, we fix the order in which prefix sums are accumulated to be the same as in $\bar{n}$; by (10) $i'$ is well defined). Next we let $\bar{n}'''$ be identical to $\bar{n}''$ except that $n_{i'}''' = n_{i'}'' + x$. In other words, $\bar{n}'''$ differs from $\bar{n}'$ by shifting $x$ from coordinate $i$ to coordinate $i'$.

We argue that incrementing $n_{i'}''$ by $x$ does not change the maximal value of $\bar{n}''$, i.e., $n_1''' > n_{i'}'''$. Our choice of $i'$, which is the smallest index so that the prefix sum over $\bar{n}''$ overtakes that over $\bar{n}$, implies that $n_1'' - n_{i'}'' > n_1 - n_{i'}$. Since $n_1'' = n_1$, it means that $n_{i'} > n_{i'}''$ (and by adding $x$ we move $n_{i'}''$ towards $n_{i'}$). Furthermore,

$$n_{i'}''' = n_{i'}'' + x \leq n_{i'} + (n_i' - n_i) = n_i' + (n_{i'} - n_i) \leq n_i' \leq n_1' = n_1''',$$

(We use $n_{i'} \leq n_i$, which is implied by $i' > i$.)

We claim that $\sum_{j=1}^{t} (n_1''' - n_j''') \leq S_t(\bar{n})$ for all $t$, and thus, via (10), $\bar{n} \succeq \bar{n}'''$. The choice of $i'$ makes the statement trivial for $t < i'$. For $t \geq i'$ the following holds:

$$\sum_{j=1}^{t} (n_1''' - n_j''') = \sum_{j=1}^{t} (n_1'' - n_j'') - x \leq \left( \sum_{j=1}^{t} (n_1' - n_j') + x \right) - x = S_t(n') \leq S_t(\bar{n}).$$

By construction $d(\bar{n}', \bar{n}''') = 1$ (the two histograms differ in two locations, in positive and negative directions, by $x \leq 1$ in each). For the same reasons $\bar{n}''' \succeq \bar{n}'$. To show that $\ell(\bar{n}, \bar{n}') > \ell(\bar{n}, \bar{n}''')$,

compare $\lceil n_j - n'_j \rceil$ and $\lceil n_j - n'''_j \rceil$ for $j = i, i'$. At $j = i$ the first term is strictly larger than the second. At $j = i'$, the inequality holds too but it may be not strict.

We may again apply the induction hypothesis to the pair $\bar{n}$ and $\bar{n}'''$, thus completing the proof of the lemma. ∎

To complete the proof of the proposition, we need to argue that the values of $q$ are also monotone in the chain constructed by the previous lemma. Concretely, we put forth

**Lemma 18.** *If $\bar{n}' \succeq \bar{n}$, $q(\bar{n}') \leq \mathrm{B_U}(q_0)$, $d(\bar{n}, \bar{n}') = 1$ and $\bar{n}^{(1)} = \bar{n}'^{(1)}$, then $q(\bar{n}) \leq q(\bar{n}')$.*

*Proof.* The fact that $d(\bar{n}, \bar{n}') = 1$ and $\bar{n} \succeq \bar{n}'$ means that there is either a single index $i$ so that $n'_i < n_i$, or there exist two indices $i$ and $j$ so that $n'_i < n_i$ and $n'_j > n_j$. The first case is immediate, since $q$ is non-decreasing in all inputs except for the largest (by C7).

Let $n'_i = n_i - x$ and $n'_j = n_j + y$, where $x, y > 0$. Since $\bar{n} \succeq \bar{n}'$, it follows that $n_i \geq n_j$ and $x > y$. Consider two cases.

**Case I:** $n'_i \geq n'_j$, i.e., removing $x$ from $n_i$ and adding $y$ to $n_j$ does not change their ordering. Let

$$\bar{n}(t) \triangleq (1 - t)\bar{n} + t \cdot \bar{n}' = [n_1, \ldots, n_i - t \cdot x, \ldots, n_j + t \cdot y, \ldots, n_m].$$

Then,

$$
\begin{aligned}
q(\bar{n}') - q(\bar{n}) = q(\bar{n}(1)) - q(\bar{n}(0)) &= \int_{t=0}^{1} (q \circ \bar{n})'(t) \, \mathrm{d}t \\
&= \int_{t=0}^{1} \left\{ -x \frac{\partial q}{\partial n_i} \bar{n}(t) + y \frac{\partial q}{\partial n_j} \bar{n}(t) \right\} \mathrm{d}t \\
&\leq 0.
\end{aligned}
$$

The last inequality follows from C7 and the facts that $x > y > 0$ and $n_i(t) > n_j(t)$. (The condition that $q(\bar{n}(t)) \leq \mathrm{B_U}(q_0)$ follows from C2 and the fact that $d(\bar{n}', \bar{n}(t)) \leq 1$.)

**Case II:** $n'_i \leq n'_j$. In this case we swap the $i$th and $j$th indices in $\bar{n}'$ by defining $\bar{n}''$ which differs from it in $\bar{n}''_i = \bar{n}'_j$ and $\bar{n}''_j = \bar{n}'_i$. By C8, $q(\bar{n}'') = q(\bar{n}')$ and, of course, $\bar{n}'' \succeq \bar{n}$ since the prefix sums remain unchanged. The benefit of doing this transformation is that we are back in Case I, where the relative order of coordinates that change between $\bar{n}$ and $\bar{n}''$ remains the same.

This concludes the proof of the lemma. ∎

Applying Lemma 17 we construct a chain of histograms between $\bar{n}$ and $\bar{n}'$, which, by Lemma 18, is non-increasing in $q(\cdot)$. Together this implies that $q(\bar{n}) \leq q(\bar{n}')$, as claimed. ∎

We apply the notion of dominance in proving the following proposition, which is used later in arguing correctness of Algorithm 4.

**Proposition 19.** *Let $\bar{n}$ be an integer-valued histogram and $d$ be a positive integer. And $q(\cdot)$ satisfies C1, C7, and C8. The following holds:*

1. *Assuming $n^{(1)} - n^{(2)} \geq 2d$, let $\bar{n}^*$ be obtained from $\bar{n}$ by decrementing $n^{(1)}$ by $d$ and incrementing $n^{(2)}$ by $d$. Then*

   $$d(\bar{n}, \bar{n}^*) = d \text{ and } q(\bar{n}^*) \geq q(\bar{n}') \text{ for any } \bar{n}' \text{ such that } d(\bar{n}, \bar{n}') = d.$$

2. *Assuming $\sum_{i=2}^{m} n^{(i)} \geq d$, let $\bar{n}^{**}$ be obtained from $\bar{n}$ by incrementing $n_1$ by $d$, and by repeatedly decrementing the histogram's* current *second highest value by one, $d$ times. Then*

   $$d(\bar{n}, \bar{n}^{**}) = d \text{ and } q(\bar{n}^{**}) \leq q(\bar{n}') \text{ for any } \bar{n}' \text{ such that } d(\bar{n}, \bar{n}') = d.$$

*Proof.* Towards proving the claims, we argue that $\bar{n}^*$ and $\bar{n}^{**}$ are, respectively, the minimal and the maximal elements in the histogram dominance order (Definition 15) in the set of histograms at distance $d$ from $\bar{n}$. By Proposition 16 the claims follow.

1. Take any histogram $\bar{n}'$ at distance $d$ from $\bar{n}$. Our goal is to prove that $\bar{n}' \succeq \bar{n}^*$. Recall the definition of the distance $d(\cdot, \cdot)$ between two histograms $d(\bar{n}, \bar{n}') = \max \left\{ \sum_{i:n_i > n'_i} \lceil n_i - n'_i \rceil, \sum_{i:n_i < n'_i} \lceil n'_i - n_i \rceil \right\}$. If the distance is bounded by $d$, it means, in particular, that

$$n'^{(1)} \geq n^{(1)} - d \text{ and}$$

$$\sum_{j=2}^{i} n'^{(j)} \leq \sum_{j=2}^{i} n^{(j)} + d \quad \text{for all } i > 2.$$

That lets us bound the prefix sums of $\bar{n}'$ as follows:

$$S_i(\bar{n}') = \sum_{j=2}^{i} (n'^{(1)} - n'^{(j)}) = (i - 1) \cdot n'^{(1)} - \sum_{j=2}^{i} n'^{(j)}$$

$$\geq (i - 1) \cdot (n^{(1)} - d) - \left( \sum_{j=2}^{i} n^{(j)} + d \right) = S_i(\bar{n}^*).$$

We demonstrated that $\bar{n}' \succeq \bar{n}^*$, which, by Proposition 16, implies that $q(\bar{n}') \leq q(\bar{n}^*)$. Together with the immediate $d(\bar{n}, \bar{n}^*) = d$ we prove the claim.

2. Assume wlog that $\bar{n}$ is sorted in the descending order. Define the following value that depends on $\bar{n}$ and $d$:

$$u \triangleq \min \left\{ x \in \mathbb{N} : \sum_{i:i>1, n_i \geq x} n_i - x \leq d \right\}.$$

The constant $u$ is the smallest such $x$ so that the total mass that can be shaved from elements of $\bar{n}$ above $x$ (excluding $n_1$) is at most $d$.

We give the following equivalent definition of $\bar{n}^{**}$:

$$n_i^{**} = \begin{cases} n_1 + d & \text{if } i = 1, \\ u & \text{if } i > 1 \text{ and } n_i \geq u, \\ n_i & \text{otherwise.} \end{cases}$$

Fix any $i \in [m]$ and any histogram $\bar{n}'$ at distance $d$ from $\bar{n}$. Our goal is to prove that $S_i(\bar{n}^{**}) \geq S_i(\bar{n}')$ and thus $\bar{n}^{**} \succeq \bar{n}'$. Assume the opposite and take largest $i$ such that $S_i(\bar{n}^{**}) < S_i(\bar{n}')$.

We may assume that $n'^{(1)} = n_1^{**} = n_1 + d$. Consider the following cases.

**Case I.** If $n^{**(i)} < u$, the contradiction follows from

$$S_i(\bar{n}') = \sum_{j=2}^{i} (n'^{(1)} - n'^{(j)}) = \sum_{j=2}^{i} \left( (n'^{(1)} - n^{(1)}) + (n^{(1)} - n^{(j)}) + (n^{(j)} - n'^{(j)}) \right)$$

$$\leq (i - 1)d + S_i(\bar{n}) + d = S_i(\bar{n}^{**}).$$

The last equality is due to the fact that all differences between $\bar{n}$ and $\bar{n}^{**}$ are confined to the indices that are less than $i$.

**Case II.** If $n^{**(i)} = u$ and $n'^{(i)} \geq u$, the contradiction with $S_i(\bar{n}^{**}) < S_i(\bar{n}')$ follows immediately from

$$S_i(\bar{n}') = \sum_{j=2}^{i} (n'^{(1)} - n'^{(j)}) \leq (i - 1)(n'^{(1)} - u) = S_i(\bar{n}^{**}).$$

**Case III.** Finally, consider the case when $n^{**(i)} = u$ and $v \triangleq n'^{(i)} < u$. Since $i$ is the largest such that $S_i(\bar{n}^{**}) < S_i(\bar{n}')$, it means that $n^{**(i+1)} < n'^{(i+1)} \leq v < u = n^{**(i)}$

and thus $n^{**(i)} - n^{**(i+1)} \geq 2$ (we rely on the fact that the histograms are integer-valued). It implies that all differences between $\bar{n}$ and $\bar{n}^{**}$ are confined to the indices in $[1, i]$. Then,

$$
\begin{aligned}
S_i(\bar{n}^{**}) - S_i(\bar{n}') &\geq \sum_{j=2}^{i}(n_1^{**} - n_j^{**}) - \sum_{j=2}^{i}(n_1' - n_j') && \text{(by (10))} \\
&= \sum_{j=2}^{i}\left((n_j - n_j^{**}) + (n_j' - n_j)\right) && \text{(since } n_1^{**} = n_1') \\
&\geq d - d(\bar{n}, \bar{n}') \\
&\geq 0,
\end{aligned}
$$

which contradicts the assumption that $S_i(\bar{n}^{**}) < S_i(\bar{n}')$.

■

We may now state and prove the main result of this section.

**Theorem 20.** *Assume that $q(\cdot)$ satisfies conditions C1–C8 and $\bar{n}$ is an integer-valued histogram. Then the following two claims are true:*

1. *Algorithm 4 computes $\max_{\bar{n}':\text{dist}(\bar{n},\bar{n}')\leq d} \tilde{\text{LS}}(\bar{n}')$ .*

2. *Algorithm 5 computes $\text{SS}_\beta(\bar{n})$, which is a $\beta$-smooth upper bound on smooth sensitivity of $\boldsymbol{\beta}(q(\cdot))$.*

*Proof.* **Claim 1.** Recall that $q_1 = \text{B}_\text{L}(q_0)$, and therefore, by Proposition 14 the function $\tilde{\text{LS}}(q)$ is non-decreasing in $[0, q_1]$, constant in $[q_1, q_0]$, and non-increasing in $[q_0, 1]$. It means, in particular, that to maximize $\tilde{\text{LS}}(q(\bar{n}'))$ over histograms satisfying $d(\bar{n}, \bar{n}') = d$, it suffices to consider the following cases.

If $\tilde{\text{LS}}(q(\bar{n})) < q_1$, then higher values of $\tilde{\text{LS}}(\cdot)$ may be attained only by histograms with higher values of $q$. Proposition 19 enables us to efficiently find a histogram $\bar{n}^*$ with the highest $q$ at distance $d$, or conclude that we may reach the plateau by making the two highest histogram entries be equal.

If $q_1 \leq \tilde{\text{LS}}(q(\bar{n})) \leq q_0$, it means that $\tilde{\text{LS}}(q(n))$ is already as high as it can be.

If $q_0 < \tilde{\text{LS}}(q(\bar{n}))$, then, according to Proposition 14, higher values of $\tilde{\text{LS}}(\cdot)$ can be achieved by histograms with smaller values of $q$, which we explore using the procedure outlined by Proposition 19. The stopping condition—when the plateau is reached—happens when $q$ becomes smaller than $q_0$.

**Claim 2.** The second claim follows from the specification of Algorithm 5 and the first claim. ■

### B.4 GNMAX SATISFIES CONDITIONS C1–C8

The previous sections laid down a framework for computing smooth sensitivity of a randomized aggregator mechanism: defining functions $q(\cdot)$, $\text{B}_\text{U}(\cdot)$, $\text{B}_\text{L}(\cdot)$, verifying that they satisfy conditions C1–C8, and applying Theorem 20, which asserts correctness of Algorithm 5. In this section we instantiate this framework for the GNMax mechanism.

### B.4.1 Conditions C1–C4, C7 and C8

**Defining $q$ and conditions C1, C7, and C8.** Following Proposition 7, we define $q \colon \mathbb{R}_{\geq 0}^m \to [0, 1]$ for a GNMax mechanism parameterized with $\sigma$ as:

$$q(\bar{n}) \triangleq \min \left\{ \sum_{i \neq i^*} \mathbf{Pr}(Z_i - Z_{i^*} \geq n_{i^*} - n_i), 1 \right\}$$

$$= \min \left\{ \sum_{i \neq i^*} \frac{1}{2} \left( 1 - \operatorname{erf} \left( \frac{n_{i^*} - n_i}{2\sigma} \right) \right), 1 \right\}$$

$$= \min \left\{ \sum_{i \neq i^*} \frac{1}{2} \operatorname{erfc} \left( \frac{n_{i^*} - n_i}{2\sigma} \right), 1 \right\},$$

where $i^*$ is the histogram $\bar{n}$'s highest coordinate, i.e., $n_{i^*} \geq n_i$ for all $i$ (if there are multiple highest, let $i^*$ be any of them). Recall that erf is the error function, and erfc is the complement error function.

Proposition 7 demonstrates that $q(\bar{n})$ bounds from above the probability that GNMax outputs anything but the highest coordinate of the histogram.

Conditions C1, C7, and C8 follow by simple calculus ($q_0$, defined below, is at most 0.5).

**Functions $B_L$, $B_U$, and conditions C2–C4.** Recall that the functions $B_L$ and $B_U$ are defined in Appendix B as follows:

$$B_U(q) \triangleq \min \left\{ \frac{m-1}{2} \operatorname{erfc} \left( \operatorname{erfc}^{-1} \left( \frac{2q}{m-1} \right) - \frac{1}{\sigma} \right), 1 \right\},$$

$$B_L(q) \triangleq \frac{m-1}{2} \operatorname{erfc} \left( \operatorname{erfc}^{-1} \left( \frac{2q}{m-1} \right) + \frac{1}{\sigma} \right),$$

**Proposition 21** (Condition C2). *For any neighbor $\bar{n}'$ of $\bar{n}$, i.e., $d(\bar{n}', \bar{n}) = 1$, the following bounds hold:*

$$B_L(q(\bar{n})) \leq q(\bar{n}') \leq B_U(q(\bar{n})).$$

*Proof.* Assume wlog that $i^* = 1$. Let $x_i \triangleq n_1 - n_i$ and $q_i \triangleq \operatorname{erfc}(x_i/2\sigma)/2$, and similarly define $x_i'$ for $\bar{n}'$. Observe that $|x_i - x_i'| \leq 2$, which, by monotonicity of erfc, implies that

$$\frac{1}{2} \operatorname{erfc} \left( \frac{x_i + 2}{2\sigma} \right) \leq q_i(\bar{n}') \leq \frac{1}{2} \operatorname{erfc} \left( \frac{x_i - 2}{2\sigma} \right).$$

Thus

$$\frac{1}{2} \sum_{i > 1} \operatorname{erfc} \left( \frac{x_i + 2}{2\sigma} \right) \leq q(\bar{n}') \leq \frac{1}{2} \sum_{i > 1} \operatorname{erfc} \left( \frac{x_i - 2}{2\sigma} \right).$$

(Although $i^*$ may change between $\bar{n}$ and $\bar{n}'$, the bounds still hold.)

Our first goal is to upper bound $q(\bar{n}')$ for a given value of $q(\bar{n})$. To this end we set up the following maximization problem

$$\max_{\{x_i\}} \frac{1}{2} \sum_{i > 1} \operatorname{erfc} \left( \frac{x_i - 2}{2\sigma} \right) \text{ such that } \frac{1}{2} \sum_{i > 1} \operatorname{erfc} \left( \frac{x_i}{2\sigma} \right) = q \text{ and } x_i \geq 0.$$

We may temporarily ignore the non-negative constraints, which end up being satisfied by our solution. Consider using the method of Lagrange multipliers and take a derivative in $x_i$':

$$-\exp \left( -\left( \frac{x_i - 2}{2\sigma} \right)^2 \right) + \lambda \exp \left( -\left( \frac{x_i}{2\sigma} \right)^2 \right) = 0$$

$$\Leftrightarrow \lambda = \exp \left( \frac{x_i - 1}{\sigma^2} \right).$$

Since the expression is symmetric in $i > 1$, it means that the local optima are attained at $x_2 = \cdots = x_m$ (the second derivative confirms that these are local maxima). After solving for $(m - 1)\operatorname{erfc}(x/2\sigma) = 2q$ we have

$$q(\bar{n}') \leq \frac{m-1}{2} \operatorname{erfc}\left(\operatorname{erfc}^{-1}\left(\frac{2q}{m-1}\right) - \frac{1}{\sigma}\right).$$

where $m$ is the number of classes. Similarly,

$$q(\bar{n}') \geq \frac{m-1}{2} \operatorname{erfc}\left(\operatorname{erfc}^{-1}\left(\frac{2q}{m-1}\right) + \frac{1}{\sigma}\right).$$

∎

Conditions C3, i.e., $\mathrm{B_L}(q)$ is monotonically increasing in $q$, and C4, i.e., $\mathrm{B_L}$ and $\mathrm{B_U}$ are functional inverses of each other in $[0, q_0]$ and $\mathrm{B_L}(q) \leq q \leq \mathrm{B_U}(q)$ for all $q \in [0, 1]$, follow from basic properties of $\operatorname{erfc}$. The restriction that $q \in [0, q_0]$ ensures that $\mathrm{B_U}(q)$ is strictly less than one, and the minimum in the definition of $\mathrm{B_U}(\cdot)$ simplifies to its first argument in this range.

### B.4.2 CONDITIONS C5 AND C6

Conditions C5 and C6 stipulate that the function $\boldsymbol{\beta}(q) \triangleq \boldsymbol{\beta}_\sigma(q)$ (defined in Appendix A) exhibits a specific growth pattern. Concretely, C5 states that $\boldsymbol{\beta}(q)$ is monotonically increasing for $0 \leq q \leq q_0$, and constant for $q_0 < q \leq 1$. (Additionally, we require that $\mathrm{B_U}(q_0) < 1$). Condition C6 requires that $\Delta\boldsymbol{\beta}(q) = \boldsymbol{\beta}(\mathrm{B_U}(q)) - \boldsymbol{\beta}(q)$ is non-decreasing in $[0, \mathrm{B_L}(q_0)]$.

Rather than proving these statements analytically, we check these assumptions for any fixed $\sigma$ and $\lambda$ via a combination of symbolic and numeric analyses.

More concretely, we construct symbolic expressions for $\boldsymbol{\beta}(\cdot)$ and $\Delta\boldsymbol{\beta}(\cdot)$ and (symbolically) differentiate them. We then minimize (numerically) the resulting expressions over $[0, q_0]$ and $[0, \mathrm{B_L}(q_0)]$, and verify that their minimal values are indeed non-negative.

### B.5 RÉNYI DIFFERENTIAL PRIVACY AND SMOOTH SENSITIVITY

Although the procedure for computing a smooth sensitivity bound may be quite involved (such as Algorithms 3–5), its use in a differentially private data release is straightforward. Following Nissim et al. (2007), we define an additive Gaussian mechanism where the noise distribution is scaled by $\sigma$ **and** a smooth sensitivity bound:

**Definition 22.** *Given a real-valued function $f \colon \mathcal{D} \to \mathbb{R}$ and a $\beta$-smooth sensitivity bound $\mathrm{SS}(\cdot)$, let $(\beta, \sigma)$-GNSS mechanism be*

$$\mathcal{F}_\sigma(D) \triangleq f(D) + \mathrm{SS}_\beta(D) \cdot \mathcal{N}(0, \sigma^2).$$

We claim that this mechanism satisfies Rényi differential privacy for finite orders from a certain range.

**Theorem 23.** *The $(\beta, \sigma)$-GNSS mechanism $\mathcal{F}_\sigma$ is $(\lambda, \varepsilon)$-RDP, where*

$$\varepsilon \triangleq \frac{\lambda \cdot e^{2\beta}}{\sigma^2} + \frac{\beta\lambda - 0.5\ln(1 - 2\lambda\beta)}{\lambda - 1}$$

*for all $1 < \lambda < 1/(2\beta)$.*

*Proof.* Consider two neighboring datasets $D$ and $D'$. The output distributions of the $(\beta, \sigma)$-GNSS mechanism on $D$ and $D'$ are, respectively,

$$P \triangleq f(D) + \mathrm{SS}_\beta(D) \cdot \mathcal{N}(0, \sigma^2) = \mathcal{N}(f(D), (\mathrm{SS}_\beta(D)\sigma)^2) \text{ and } Q \triangleq \mathcal{N}(f(D'), (\mathrm{SS}_\beta(D')\sigma)^2).$$

The Rényi divergence between two normal distributions can be computed in closed form (van Erven & Harremoës, 2014):

$$D_\lambda(P\|Q) = \lambda\frac{(f(D) - f(D'))^2}{2\sigma^2 s^2} + \frac{1}{1 - \lambda}\ln\frac{s}{\mathrm{SS}_\beta(D)^{1-\lambda} \cdot \mathrm{SS}_\beta(D')^\lambda}, \tag{11}$$

provided $s^2 \triangleq (1 - \lambda) \cdot \mathrm{SS}_\beta(D)^2 + \lambda \cdot \mathrm{SS}_\beta(D')^2 > 0$.

According to the definition of smooth sensitivity (Definition 12)

$$e^{-\beta} \cdot \mathrm{SS}_\beta(D) \leq \mathrm{SS}_\beta(D') \leq e^{\beta} \cdot \mathrm{SS}_\beta(D), \tag{12}$$

and

$$|f(D) - f(D')| \leq e^{\beta} \cdot \min(\mathrm{SS}_\beta(D), \mathrm{SS}_\beta(D')). \tag{13}$$

Bound (12) together with the condition that $\lambda \leq 1/(2\beta)$ implies that

$$
\begin{aligned}
s^2 = (1 - \lambda) \cdot \mathrm{SS}_\beta(D)^2 + \lambda \cdot \mathrm{SS}_\beta(D')^2 &= \mathrm{SS}_\beta(D)^2 + \lambda(\mathrm{SS}_\beta(D')^2 - \mathrm{SS}_\beta(D)^2) \\
&\geq \mathrm{SS}_\beta(D)^2(1 + \lambda(e^{-2\beta} - 1)) \geq \mathrm{SS}_\beta(D)^2(1 - 2\lambda\beta) > 0. \quad (14)
\end{aligned}
$$

The above lower bound ensures that $s^2$ is well-defined, i.e., non-negative, as required for application of (11).

Combining bounds (12)– (14), we have that

$$D_\lambda(P\|Q) \leq \frac{\lambda \cdot e^{2\beta}}{\sigma^2} + \frac{1}{1-\lambda} \ln\left\{ \frac{s}{\mathrm{SS}_\beta(D)} e^{-\lambda\beta} \right\} \leq \frac{\lambda \cdot e^{2\beta}}{\sigma^2} + \frac{\beta\lambda - 0.5\ln(1 - 2\lambda\beta)}{\lambda - 1}$$

as claimed. ∎

Note that if $\lambda \gg 1$, $\sigma \ll \lambda$, and $\beta \ll 1/(2\lambda)$, then $(\beta, \sigma)$-GNSS satisfies $(\lambda, (\lambda+1)/\sigma^2)$-RDP. Compare this with RDP analysis of the standard additive Gaussian mechanism, which satisfies $(\lambda, \lambda/\sigma^2)$-RDP. The difference is that GNSS scales noise in proportion to *smooth sensitivity*, which is no larger and can be much smaller than global sensitivity.

### B.6   PUTTING IT ALL TOGETHER: APPLYING SMOOTH SENSITIVITY

Recall our initial motivation for the smooth sensitivity analysis: enabling privacy-preserving release of data-dependent privacy guarantees. Indeed, these guarantees vary greatly between queries (see Figure 5) and are typically much smaller than data-independent privacy bounds. Since data-dependent bounds may leak information about underlying data, publishing the bounds themselves requires a differentially private mechanism. As we explain shortly, smooth sensitivity analysis is a natural fit for this task.

We first consider the standard additive noise mechanism where the noise (such as Laplace or Gaussian) is calibrated to the global sensitivity of the function we would like to make differentially private. We know that Rényi differential privacy is additive for any fixed order $\lambda$, and thus the cumulative RDP cost is the sum of RDP costs of individual queries each upper bounded by a data-independent bound. Thus, it might be tempting to use the standard additive noise mechanism for sanitizing the total, but that would be a mistake.

To see why, consider a sequence of queries $\bar{n}_1, \ldots, \bar{n}_\ell$ answered by the aggregator. Their total (unsanitized) RDP cost of order $\lambda$ is $B_\sigma = \sum_{i=1}^{\ell} \beta_\sigma(q(\bar{n}_i))$. Even though $\beta_\sigma(q(\bar{n}_i)) \leq \lambda/\sigma^2$ (the data-independent bound, Proposition 8), the sensitivity of their sum is *not* $\lambda/\sigma^2$. The reason is that the (global) sensitivity is defined as the maximal difference in the function's output between two neighboring datasets $D$ and $D'$. Transitioning from $D$ to $D'$ may change one teacher's output on *all* student queries.

In contrast with the global sensitivity of $B_\sigma$ that may be quite high—particularly for the second step of the Confident GNMax aggregator—its smooth sensitivity can be extremely small. Towards computing a smooth sensitivity bound on $B_\sigma$, we prove the following theorem which defines a smooth sensitivity of the sum in terms of local sensitivities of its parts.

**Theorem 24.** *Let $f_i \colon \mathcal{D} \to \mathbb{R}$ for $1 \leq i \leq \ell$, $F(D) \triangleq \sum_{i=1}^{\ell} f_i(D)$ and $\beta > 0$. Then*

$$\mathrm{SS}(D) \triangleq \max_{d \geq 0} e^{-\beta d} \cdot \sum_{i=1}^{\ell} \max_{D':\mathrm{dist}(D,D') \leq d} \widetilde{\mathrm{LS}}_{f_i}(D'),$$

*is a $\beta$-smooth bound on $F(\cdot)$ if $\widetilde{\mathrm{LS}}_{f_i}(D')$ are upper bounds on the local sensitivity of $f_i(D')$.*

*Proof.* We need to argue that $\mathrm{SS}(\cdot)$ is $\beta$-smooth, i.e., $\mathrm{SS}(D_1) \leq e^\beta \cdot \mathrm{SS}(D_2)$ for any neighboring $D_1, D_2 \in \mathcal{D}$, and it is an upper bound on the local sensitivity of $F(D_1)$, i.e., $\mathrm{SS}(D_1) \geq |F(D_1) - F(D_2)|$.

Smoothness follows from the observation that

$$\max_{D:\mathrm{dist}(D_1,D)\leq d} \tilde{\mathrm{LS}}_{f_i}(D) \leq \max_{D:\mathrm{dist}(D_2,D)\leq d+1} \tilde{\mathrm{LS}}_{f_i}(D)$$

for all neighboring datasets $D_1$ and $D_2$ (by the triangle inequality over distances). Then

$$\mathrm{SS}(D_1) = \max_{d\geq 0} e^{-\beta d} \cdot \sum_{i=1}^{\ell} \max_{D:\mathrm{dist}(D_1,D)\leq d} \tilde{\mathrm{LS}}_{f_i}(D),$$

$$\leq \max_{d\geq 0} e^{-\beta d} \cdot \sum_{i=1}^{\ell} \max_{D:\mathrm{dist}(D_2,D)\leq d+1} \tilde{\mathrm{LS}}_{f_i}(D)$$

$$= \max_{d'\geq 1} e^{-\beta(d'-1)} \cdot \sum_{i=1}^{\ell} \max_{D:\mathrm{dist}(D_2,D)\leq d'} \tilde{\mathrm{LS}}_{f_i}(D)$$

$$\leq e^\beta \cdot \mathrm{SS}(D_2)$$

as needed for $\beta$-smoothness.

The fact that $\mathrm{SS}(\cdot)$ is an upper bound on the local sensitivity of $F(\cdot)$ is implied by the following:

$$|F(D_1) - F(D_2)| = \left| \sum_{i=1}^{\ell} f_i(D_1) - \sum_{i=1}^{\ell} f_i(D_2) \right|$$

$$\leq \sum_{i=1}^{\ell} |f_i(D_1) - f_i(D_2)|$$

$$\leq \sum_{i=1}^{\ell} \tilde{\mathrm{LS}}_{f_i}(D_1)$$

$$\leq \mathrm{SS}(D_1),$$

which concludes the proof. ∎

Applying Theorem 24 allows us to compute a smooth sensitivity of the sum more efficiently than summing up smooth sensitivities of its parts. Results below rely on this strategy.

**Empirical results.** Table 2 revisits the privacy bounds in Table 1. For all data-dependent privacy claims of the Confident GNMax aggregator we report parameters for their smooth sensitivity analysis and results of applying the GNSS mechanism for their release.

Consider the first row of the table. The MNIST dataset was partitioned among 250 teachers, each getting 200 training examples. After the teachers were individually trained, the student selected at random 640 unlabeled examples, and submitted them to the Confident GNMax aggregator with the threshold of 200, and noise parameters $\sigma_1 = 150$ and $\sigma_2 = 40$. The expected number of answered examples (those that passed the first step of Algorithm 1) is 283, and the expected Rényi differential privacy is $\varepsilon = 1.18$ at order $\lambda = 14$. This translates (via Theorem 5) to $(2.00, 10^{-5})$-differential privacy, where 2.00 is the expectation of the privacy parameter $\varepsilon$.

These costs are data-dependent and they cannot be released without further sanitization, which we handle by adding Gaussian noise scaled by the smooth sensitivity of $\varepsilon$ (the GNSS mechanism, Definition 22). At $\beta = 0.0329$ the expected value of smooth sensitivity is 0.0618. We choose $\sigma_{\mathrm{SS}} = 6.23$, which incurs, according to Theorem 23, an additional (data-independent) $(14, 0.52)$-RDP cost. Applying $(\beta, \sigma_{\mathrm{SS}})$-GNSS where $\sigma_{\mathrm{SS}} = 6.23$, we may publish differentially private estimate of the total privacy cost that consists of a fixed part—the cost of applying Confident GN-Max and GNSS—and random noise. The fixed part is $2.52 = 1.18 + 0.52 - \ln(10^{-5})/14$, and the noise is normally distributed with mean 0 and standard deviation $\sigma_{\mathrm{SS}} \cdot 0.0618 = 0.385$. We note

| Dataset | Confident GNMax parameters | DP $\mathbb{E}\left[\varepsilon\right]$ | $\delta$ | Smooth Sensitivity $\lambda$ | $\beta$ | $\mathbb{E}\left[\mathrm{SS}_\beta\right]$ | $\sigma_{\mathrm{SS}}$ | Sanitized DP $\mathbb{E}\left[\varepsilon\right]\pm$ noise |
|---|---|---|---|---|---|---|---|---|
| MNIST | $T$=200, $\sigma_1$=150, $\sigma_2$=40 | 2.00 | $10^{-5}$ | 14 | .0329 | .0618 | 6.23 | $2.52 \pm 0.385$ |
| SVHN | $T$=300, $\sigma_1$=200, $\sigma_2$=40 | 4.96 | $10^{-6}$ | 7.5 | .0533 | .0717 | 4.88 | $5.45 \pm 0.350$ |
| Adult | $T$=300, $\sigma_1$=200, $\sigma_2$=40 | 1.68 | $10^{-5}$ | 15.5 | .0310 | 0.0332 | 7.92 | $2.09 \pm 0.263$ |
| Glyph | $T$=1000, $\sigma_1$=500, $\sigma_2$=100 | 2.07 | $10^{-8}$ | 20.5 | .0205 | .0128 | 11.9 | $2.29 \pm 0.152$ |
| | Two-round interactive | 0.837 | $10^{-8}$ | 50 | .009 | .00278 | 26.4 | $1.00 \pm .081$ |
| | | | | | .008 | .00088 | 38.7 | |

Table 2: **Privacy-preserving reporting of privacy costs.** The table augments Table 1 by including smooth sensitivity analysis of the total privacy cost. The expectations are taken over the student's queries and outcomes of the first step of the Confident GNMax aggregator. Order $\lambda$, smooth sensitivity parameter $\beta$, $\sigma_{\mathrm{SS}}$ are parameters of the GNSS mechanism (Section B.5). The final column sums up the data-dependent cost $\varepsilon$, the cost of applying GNSS (Theorem 23), and the standard deviation of Gaussian noise calibrated to smooth sensitivity (the product of $\mathbb{E}\left[\mathrm{SS}_\beta\right]$ and $\sigma_{\mathrm{SS}}$).

that, in contrast with the standard additive noise, one cannot publish its standard deviation without going through additional privacy analysis.

Some of these constants were optimally chosen (via grid search or analytically) given full view of data, and thus provide a somewhat optimistic view of how this pipeline might perform in practice. For example, $\sigma_{\mathrm{SS}}$ in Table 2 were selected to minimize the total privacy cost plus two standard deviation of the noise.

The following rules of thumb may replace these laborious and privacy-revealing tuning procedures in typical use cases. The privacy parameter $\delta$ must be less than the inverse of the number of training examples. Giving a target $\varepsilon$, the order $\lambda$ can be chosen so that $\log(1/\delta) \approx (\lambda - 1)\varepsilon/2$, i.e., the cost of the $\delta$ contribution in Theorem 5 be roughly half of the total. The $\beta$-smoothness parameter can be set to $0.4/\lambda$, from which smooth sensitivity $\mathrm{SS}_\beta$ can be estimated. The final parameter $\sigma_{\mathrm{SS}}$ can be reasonably chosen between $2 \cdot \sqrt{(\lambda + 1)/\varepsilon}$ and $4 \cdot \sqrt{(\lambda + 1)/\varepsilon}$ (ensuring that the first, dominant component, of the cost of the GNSS mechanism given by Theorem 23 is between $\varepsilon/16$ and $\varepsilon/4$).

