# OpenReview forum: "Scalable Private Learning with PATE"
_ICLR.cc/2018/Conference — Accept (Poster)_

### Official Review · AnonReviewer1 · 2017-11-27
**Novel techniques to improve private learning with PATE**

**Rating:** 6
**Confidence:** 1

**Review:**

The paper proposes novel techniques for private learning with PATE framework. Two key ideas in the paper include the use of Gaussian noise for the aggregation mechanism in PATE instead of Laplace noise and selective answering strategy by teacher ensemble. In the experiments, the efficacy of the proposed techniques has been demonstrated. I am not familiar with privacy learning but it is interesting to see that more concentrated distribution (Gaussian) and clever aggregators provide better utility-privacy tradeoff.

1. As for noise distribution, I am wondering if the variance of the distribution also plays a role to keep good utility-privacy trade-off. It would be great to discuss and show experimental results for utility-privacy tradeoff with different variances of Laplace and Gaussian noise.

2. It would be great to have an intuitive explanation about differential privacy and selective aggregation mechanisms with examples.

3. It would be great if there is an explanation about the privacy cost for selective aggregation. Intuitively, if teacher ensemble does not answer, it seems that it would reveal the fact that teachers do not agree, and thus spend some privacy cost.

---

> ### Author Response · Authors · 2017-12-15
> **Response to review 1**
>
> We thank the reviewer for their feedback. Below are answers for each of the three points included in your feedback.
>
> 1. You are right that the variance of the distribution plays a fundamental role in the utility-privacy tradeoff. Roughly speaking, larger noise variances typically yield stronger privacy guarantees but reduce the utility of the aggregated label. We updated Figure 2 to illustrate this relationship, separating the effects of the shape of the noise distribution and the number of teachers.
>
> The left chart of Figure 2 plots the utility-privacy tradeoff for the Laplace (prior work) and the Gaussian (ours) aggregation mechanisms. The measurement points are labelled with the standard deviation of the noise (sigma) which ranges from 5 to 200. As intuition suggests, the accuracy decreases with the variance of the noise. The privacy cost cannot be measured directly (as it involves considering all possible counterfactuals). Rather, it is computed according to Theorem 2, which is a significant technical contribution of this paper. For some ranges of parameters, privacy costs are a non-monotone function of sigma, which is discussed in the end of Section 4.2.
>
> 2. We acknowledge the reviewer's comment about giving an intuitive overview of how we achieve differential privacy. The intuitive privacy guarantee of PATE remains the same as that presented in the original PATE paper by Papernot et al.: Partitioning training data ensures that the presence or absence of a single training data point affects at most one teacher’s vote. By adding noise to the teacher votes, we control the impact of a single teacher vote in the final outcome of the aggregation mechanism, resulting from the plurality of votes in the teacher ensemble. In fact, precisely bounding this impact, in a tighter way, requires the use of data-dependent analysis. The large variability between queries’ privacy costs motivates the Confident Aggregator, which minimizes total privacy budget by being selective in the student queries it chooses to answer.
>
> To see why the Confident Aggregator is useful, here are a couple of illustrations of cheap and expensive queries in terms of their privacy cost. Consider our ensembles of 5000 teachers with the following votes across classes (4900, 100, 0, … 0). You can see that there is an overwhelming consensus and after adding noise, the chance that Class 1 is output by the ensemble is still very high. The resulting privacy cost (at Gaussian noise with stdev 100) is 3.6e-249.
>
> However, if there’s poor consensus amongst the teachers, with votes, say, (2500, 2400, 100, … ) i.e., the ensemble is rather confused between Classes 1 and 2, then the resulting privacy cost is 0.0025. It is also easy to see intuitively for our choices of thresholds, why the first example would almost surely be selected but the second example would likely not pass the threshold check.
>
> Unfortunately due to space constraints, we could not provide detailed intuition on these aspects. Some of them follow from the original PATE paper, and we try to give some insight into how the Confident Aggregator works through the experiment in Section 4.4.2. In particular, Figure 6 shows how queries along the lines of the second example above are eliminated.
>
> 3. You are correct: private information may be revealed when the teacher ensemble does not answer because it indicates that teachers do not agree on the prediction. This is why our selective aggregation mechanisms choose in a privacy-preserving way queries that will be answered. This consideration motivates the design of the condition found at line 1 of Algorithm 1 and line 2 of Algorithm 2 in the submission draft, which both add Gaussian noise with variance $\sigma_1^2$ before applying the maximum operator and comparing the result to the predefined threshold T. This ensures that the teacher consensus is checked in differentially private manner. In fact, _most_ of the total privacy budget is committed to selecting the set of queries whose answers are going to be revealed. We thank you for bringing this to our attention and updated the introduction of Section 3, as well as the captions of Algorithms 1 and 2 to emphasize this important consideration.

---

### Official Review · AnonReviewer2 · 2017-11-27
**This work investigates scalable applications of PATE**

**Rating:** 6
**Confidence:** 4

**Review:**

Summary:
In this work, PATE, an approach for learning with privacy,  is modified to scale its application to real-world data sets. This is done by leveraging the synergy between privacy and utility, to make better use of the privacy budget spent when transferring knowledge from teachers to the student. Two aggregation mechanisms are introduced for this reason.  It is demonstrated that sampling from a Gaussian distribution (instead from a Laplacian distribution) facilitates the aggregation of teacher votes in tasks with large number of output classes.

on the positive side:

Having scalable models is important, especially models that can be applied to data with privacy concerns. The extension of an approach for learning with privacy to make it scalable is of merit. The paper is well written, and the idea of the model is clear.


on the negative side:

In the introduction, the authors introduce the problem by the importance of privacy issues in medical and health care data. This is for sure an important topic. However, in the following paper, the model is applied no neither medical nor healthcare data. The authors mention that the original model PATE was applied to medical record and census data with the UCI diabetes and adult data set. I personally would prefer to see the proposed model applied to this kind of data sets as well.

minor comments:

Figure 2, legend needs to be outside the Figure, in the current Figure a lot is covered by the legend

---

> ### Author Response · Authors · 2017-12-15
> **Response to review 2**
>
> We thank you for your feedback, in particular for bringing to our attention the possible improvements to our experimental setup with respect to the datasets considered. In our submission draft, we chose to focus on the Glyph dataset because it presented challenges like class imbalance and mislabeled data. However, we agree that in order to facilitate a comparison with the original PATE publication, it is important to include results on other datasets such as the UCI Adult and Diabetes datasets. As such, we used the resources made publicly available by the authors of the original PATE publication to reproduce their results and measure the performance of our refined aggregation mechanisms on these two datasets.
>
> We also ran our experiments on the Glyph dataset with the aggregator used by Papernot et al. in the original PATE publication to provide an additional point of comparison.
>
> These additional results are now included in our last submission revision and are summarized in Figure 5. We show that we compare favorably on all of the datasets and models: we either improve student accuracy, strengthen privacy guarantees, or both simultaneously.
>
> We also followed your suggestion of making the legend of Figure 2 less intrusive by splitting the Figure into two, reducing the amount of information hidden by the legend.

---

### Official Review · AnonReviewer3 · 2017-12-04
**Clarification needed for data-dependent privacy guarantee.**

**Rating:** 7
**Confidence:** 3

**Review:**

This paper considers the problem of private learning and uses the PATE framework to achieve differential privacy. The dataset is partitioned and multiple learning algorithms produce so-called teacher classifiers. The labels produced by the teachers are aggregated in a differentially private manner and the aggregated labels are then used to train a student classifier, which forms the final output. The novelty of this work is a refined aggregation process, which is improved in three ways:
a) Gaussian instead of Laplace noise is used to achieve differential privacy.
b) Queries to the aggregator are "filtered" so that the limited privacy budget is only expended on queries where the teachers are confident and the student is uncertain or wrong.
c) A data-dependent privacy analysis is used to attain sharper bounds on the privacy loss with each query.

I think this is a nice modular framework form private learning, with significant refinements relative to previous work that make the algorithm more practical. On this basis, I think the paper should be accepted. However, I think some clarification is needed with regard to item c above:

Theorem 2 gives a data-dependent privacy guarantee. That is, if there is one label backed by a clear majority of teachers, then the privacy loss (as measured by Renyi divergence) is low. This data-dependent privacy guarantee is likely to be much tighter than the data-independent guarantee.
However, since the privacy guarantee now depends on the data, it is itself sensitive information. How is this issue resolved? If the final privacy guarantee is data-dependent, then this is very different to the way differential privacy is usually applied. This would resemble the "privacy odometer" setting of Rogers-Roth-Ullman-Vadhan [ https://arxiv.org/abs/1605.08294 ].
Another way to resolve this would be to have an output-dependent privacy guarantee. That is, the privacy guarantee would depend only on public information, rather than the private data. The widely-used "sparse vector" technique [ http://www.cis.upenn.edu/~aaroth/Papers/privacybook.pdf#page=59 ] does this.
In any case, this is an important issue that needs to be clarified, as it is not clear to me how this is resolved.

The algorithm in this work is similar to the so-called median mechanism [ https://www.cis.upenn.edu/~aaroth/Papers/onlineprivacy.pdf ] and private multiplicative weights [ http://mrtz.org/papers/HR10mult.pdf ]. These works also involve a "student" being trained using sensitive data with queries being answered in a differentially private manner. And, in particular, these works also filter out uninformative queries using the sparse vector technique. It would be helpful to add a comparison.

---

> ### Author Response · Authors · 2017-12-23
> **Response to Reviewer3**
>
> We thank the reviewer for their feedback.
>
> Before we address specific points raised by the reviewer, we offer a short summary of updates to the submission. The introduction was substantially revised and now includes Figure 1 that illustrates our improvements over the original PATE work. Table 1 includes updated parameters that dominate previously reported state-of-the-art utility and privacy on standard datasets. Figures 4 and 5 illustrate savings to privacy costs due to selective answers. The Appendix was expanded to include discussion of smooth sensitivity.
>
> The reviewer correctly notes that the privacy guarantees depend on sensitive information and this should be accounted for. In Section 2.4, we briefly mention that we handle this using the smooth sensitivity framework (as in the original PATE paper) and in our revised draft of the submission, we provide a detailed overview of our smooth sensitivity analysis in Appendix C.3.
>
> The reported numbers in the rest of our work are without this smooth sensitivity noise added to the privacy cost, because we find that the smooth sensitivity is small enough that adding noise to sanitize the privacy cost itself does not have a significant impact on the value of the privacy cost for the cases reported. We will release the code to calculate the smooth sensitivity along with the next version of our paper.
>
> While the work of Rogers et al. deals with related issues, it operates in a different setting. There, the privacy cost is input-independent, but is a function of the output and of when the mechanism is stopped. While it would be useful to have an odometer-like scheme where we can run the mechanism until a certain privacy budget is consumed, we find that running for a fixed number of steps and estimating the privacy spent using the smooth sensitivity framework largely suffices for the current work.
>
> The sparse-vector technique, the works on private multiplicative weight (PMW) and the median mechanism are related to our work and we have added more discussion on them in the related work section (Appendix B). Both PMW and the Median mechanism can be thought of as algorithms that first privately select a small number of queries to answer using the database, and then answer them using a differentially private mechanism. In PMW and Median, the selection is drastic and prunes out all but a tiny fraction of queries, and the sparse vector technique allows one to analyze this step. In our case, the selection ends up choosing (or discarding) a non-trivial fraction of the queries (between 30% and 90% for parameter settings reported in the submission). We explored using the sparse vector technique for this part of the analysis and it did not lead to any reduction in the privacy cost: while we pay only for the selected queries, the additional constant factors in the sparse vector technique wash out this benefit.
>
> For the second part of actually answering the queries, PMW and Median do a traditional data-independent privacy analysis. In our setting, using a data-independent privacy analysis in the second step would require a lot more noise than the learning process can tolerate, and we benefit greatly from using a data-dependent privacy analysis. The selection not only cuts down the number of queries that are answered, but more importantly, selects for queries that are cheap to answer from the privacy point of view. In summary: In PMW, the goal of the selection is to reduce number of answered queries from Q to log Q, and one does a data-independent privacy analysis for answering those. In our case the goal of filtering is to select a constant fraction of queries that will have a clear majority, so that the data-dependent privacy cost is small.

---

### Decision · Program_Chairs · 2018-01-29
**ICLR 2018 Conference Acceptance Decision**

**Decision:**

Accept (Poster)

**Comment:**

This paper extends last year's paper on PATE to large-scale, real-world datasets.  The model works by training multiple "teacher" models -- one per dataset, where a dataset might be for example, one user's data -- and then distilling those models into a student model. The teachers are all trained on disjoint data. Differential privacy is guaranteed by aggregating the teacher responses with added noise.  The paper shows improved teacher consensus by adding more concentrated noise and allowing the teacher to simply not respond to a student query.  The new results beat the old results convincingly on a variety of measures.

Quality and Clarity: The reviewers and I thought the paper was well written.

Originality: In some sense this work is incremental, extending and improving the existing PATE framework.  However, the extensions and new analysis are non-trivial and the results are good.

PROS:
1. Well written though difficult in places for somebody like myself who is not involved in this area.
2. Much improved scalability to real datasets
3. Good theoretical analysis supporting the extensions.
4. Comparison to related work (with a new comparison to UCI medical datasets used in the original paper and better results)

CONS:
1. Perhaps a little dense for the non-expert